# LOSSY IMAGE COMPRESSION
# WITH CONDITIONAL DIFFUSION MODELS

## ABSTRACT

Denoising diffusion models have recently marked a milestone in high-quality image generation. One may thus wonder if they are suitable for neural image compression. This paper outlines an end-to-end optimized image compression framework based on a conditional diffusion model, drawing on the transform-coding paradigm. Besides the latent variables inherent to the diffusion process, this paper introduces an additional discrete "content" latent variable to condition the denoising process. This variable is equipped with a hierarchical prior for entropy coding. The remaining "texture" latent variables characterizing the diffusion process are synthesized (either stochastically or deterministically) at decoding time. We furthermore show that the performance can be tuned toward perceptual metrics of interest. Our extensive experiments involving five datasets and sixteen image quality assessment metrics show that our approach not only compares favorably in rate-perceptual quality but also shows close distortion performance with state-of-the-art models.

## 1 INTRODUCTION

With visual media vastly dominating consumer internet traffic, developing new efficient codecs for images and videos has become evermore crucial (Cisco, 2017). The past few years have shown considerable progress in deep learning-based image codecs that have outperformed classical codecs in terms of the inherent tradeoff between rate (expected file size) and distortion (quality loss) (Ballé et al., 2018; Minnen et al., 2018; Minnen & Singh, 2020; Zhu et al., 2021; Yang et al., 2020; Cheng et al., 2020; Yang et al., 2022b). Recent research promises even more compression gains upon optimizing for perceptual quality, i.e., increasing the tolerance for imperceivable distortion for the benefit of lower rates (Blau & Michaeli, 2019). For example, recent works involving adversarial losses (Agustsson et al., 2019; Mentzer et al., 2020) show good perceptual quality at low bitrates.

Most state-of-the-art learned codecs currently rely on the transform coding paradigm and involve hierarchical "compressive" variational autoencoders (Ballé et al., 2018; Minnen et al., 2018; Minnen & Singh, 2020). These models simultaneously transform the data into a lower dimensional latent space and use a learned prior model for entropy-coding the latent representations into short bit strings. Using either Gaussian or Laplacian decoders, these models directly optimize for low MSE/MAE distortion performance. Given the increasing focus on perceptual performance over distortion, and VAEs suffer from mode averaging behavior inducing blurriness (Zhao et al., 2017), one may wonder if better perceptual results can be expected by replacing the Gaussian decoder with a more expressive conditional generative model.

This paper proposes to relax the typical requirement of Gaussian (or Laplacian) decoders in compression setups and presents a more expressive generative model instead: a conditional diffusion model. Diffusion models have achieved remarkable results on high-quality image generation tasks (Ho et al., 2020; Song et al., 2021b;a). By hybridizing hierarchical compressive VAEs (Ballé et al., 2018) with conditional diffusion models, we create a novel deep generative model with promising properties for perceptual image compression. This approach is related to but distinct from the recently proposed Diff-AEs (Preechakul et al., 2022), which are neither variational (as needed for entropy coding) nor tailored to the demands of image compression.

We evaluate our new compression model on five datasets and investigate a total of 16 different metrics, ranging from distortion metrics, perceptual reference metrics, and no-reference perceptual

metrics. We find that the approach is comparable with the best available compression models while showing more consistent behavior across the different tasks. We also show that making the decoder more stochastic vs. deterministic will decrease over-smoothing while degrading distortion, showing once more that perceptual quality is distinct from good reconstruction (Blau & Michaeli, 2019).

In sum, our contributions are as follows:

- We propose the first transform-coding-based lossy compression scheme using diffusion models. The approach uses a VAE-style encoder to map images onto a contextual latent variable; this latent variable is then fed as context into a diffusion model for reconstructing the data. The approach can be modified to enhance several perceptual metrics of interest.

- We derive our model's loss function systematically from a variational lower bound to the data log-likelihood. The resulting distortion term is distinct from traditional VAEs and is better suited for modeling the residual noise than a conditional Gaussian distribution.

- We provide substantial empirical evidence that a variant of our approach is, in many cases, better than the state-of-the-art in term of perceptual quality. Our base model also shows on-par rate-distortion performance with two MSE-optimized baselines. To this end, we considered five test sets, three baseline models (Wang et al., 2022; Mentzer et al., 2020; Cheng et al., 2020), and 16 image quality assessment metrics (classical and neural).

## 2 RELATED WORK

We discuss related works on *Lossy Compression*, *Compression For Realism* and *Diffusion Models*.

**Lossy Image Compression** The widely-established classical codecs such as JPEG (Wallace, 1991), BPG (Bellard, 2018), WEBP (Google, 2022) have recently been challenged by end-to-end learned codecs (Ballé et al., 2018; Minnen et al., 2018; Minnen & Singh, 2020; Yang et al., 2020; Cheng et al., 2020; Zhu et al., 2021). These methods typically draw on the non-linear transform coding paradigm as realized by hierarchical VAEs. Usually, neural codecs are optimized to simultaneously minimize rate and *distortion* metrics, such as mean squared error or structural similarity.

**Compression For Realism** In contrast to neural compression approaches targeting traditional metrics, some recent works have explored compression models to enhance *realism* (Agustsson et al., 2019; Mentzer et al., 2020; Tschannen et al., 2018). A theoretical background for these approaches was provided by Blau & Michaeli (2019); Zhang et al. (2021), who considered optimizing the autoencoder-based compression model with additional distortion terms based on neural metrics (e.g. LPIPS (Zhang et al., 2018a)) or adversarial losses (Goodfellow et al., 2014; Rippel & Bourdev, 2017). Since GAN training introduces a variety of instabilities, successful deployment of these methods requires a variety of design choices.

**Diffusion Models** Probabilistic diffusion models showed impressive performance on image generation tasks, with perceptual qualities comparable to those of highly-tuned GANs while maintaining stable training (Song & Ermon, 2019; Ho et al., 2020; Song et al., 2021b; Song & Ermon, 2019; Kingma et al., 2021; Yang et al., 2022a; Ho et al., 2022b; Saharia et al., 2022; Preechakul et al., 2022). Popular recent diffusion models include Dall-E2 (Ramesh et al., 2022) and Stable-Diffusion (Rombach et al., 2022). Some works also proposed diffusion models for compression. Hoogeboom et al. (2021) evaluated an autoregressive diffusion model (ADM) on a lossless compression task. Besides the difference between lossy and lossless compression, the model is only tested on low-resolution CIFAR-10 (Krizhevsky et al., 2009) dataset. In concurrent work, Theis et al. (2022) proposed a diffusion model for lossy compression, using a generic unconditional diffusion model that can communicate Gaussian samples, but there is currently no practical method that can reduce its extensive computational cost without restrictive assumption or additional coding costs (Li & El Gamal, 2018; Flamich et al., 2020; 2022; Theis & Ahmed, 2022).

## 3 METHOD

We review diffusion models and neural compression methods and then discuss our model design.

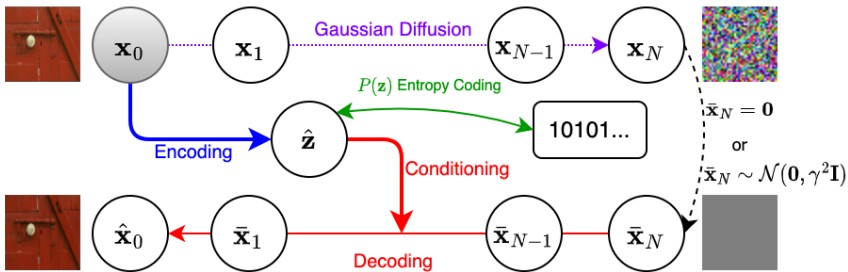

Figure 1: Overview of our proposed compression architecture. A discrete "content" latent variable $\hat{\mathbf{z}}$ contains information about the image. Upon decoding, this variable us used for conditioning a denoising diffusion process. The involved "texture" latent variables $\bar{\mathbf{x}}_{1:N}$ are synthesized on the fly.

### 3.1 BACKGROUND

**Denoising diffusion models** are hierarchical latent variable models that generate data by a sequence of iterative stochastic denoising steps (Sohl-Dickstein et al., 2015; Ho et al., 2020; Song et al., 2021a; Song & Ermon, 2019). These models describe a joint distribution over data $\mathbf{x}_0$ and latent variables $\mathbf{x}_{1:N}$ such that $p_\theta(\mathbf{x}_0) = \int p_\theta(\mathbf{x}_{0:N})d\mathbf{x}_{1:N}$. While a diffusion process (denoted by $q$) incrementally *destroys* structure, its reverse process $p_\theta$ *generates* structure. Both processes involve Markovian dynamics between a sequence of transitional steps (denoted by $n$), where

$$q(\mathbf{x}_n|\mathbf{x}_{n-1}) = \mathcal{N}(\mathbf{x}_n|\sqrt{1-\beta_n}\mathbf{x}_{n-1}, \beta_n\mathbf{I}); \quad p_\theta(\mathbf{x}_{n-1}|\mathbf{x}_n) = \mathcal{N}(\mathbf{x}_{n-1}|M_\theta(\mathbf{x}_n, n), \beta_n\mathbf{I}). \quad (1)$$

The variance schedule $\beta_n \in (0, 1)$ can be either fixed or learned; besides it, the diffusion process is parameter-free. The denoising process predicts the posterior mean from the diffusion process and is parameterized by a neural network $M_\theta(\mathbf{x}_n, n)$.

Denoising Diffusion Probabilistic Model (DDPM) (Ho et al., 2020) showed a tractable objective function for training the reverse process. A simplified version of their objective resulted in the following *noise parameterization*, where one seeks to predict the noise $\epsilon$ used to perturb a particular image $\mathbf{x}_0$ from the noisy image $\mathbf{x}_n$ at noise level $n$:

$$L(\theta, \mathbf{x}_0) = \mathbb{E}_{n,\epsilon}||\epsilon - \epsilon_\theta(\mathbf{x}_n(\mathbf{x}_0), n)||^2. \quad (2)$$

Above, $n \sim \text{Unif}\{1, ..., N\}$, $\epsilon \sim \mathcal{N}(\mathbf{0}, \mathbf{I})$, $\mathbf{x}_n(\mathbf{x}_0) = \sqrt{\alpha_n}\mathbf{x}_0 + \sqrt{1-\alpha_n}\epsilon$, and $\alpha_n = \prod_{i=1}^{n}(1-\beta_i)$. At test time, data can be generated by ancestral sampling using Langevin dynamics. Alternatively, Song et al. (2021a) proposed the Denoising Diffusion Implicit Model (DDIM) that follows a deterministic generation procedure after an initial stochastic draw from the prior. Our paper uses the DDIM scheme at test time, see Section 3.2 for details.

**Neural image compression** seeks to outperform traditional image codecs by machine-learned models. Our approach draws on the transform-coding-based neural image compression approach (Theis et al., 2017; Ballé et al., 2018; Minnen et al., 2018), where the data are non-linearly transformed into a latent space, and subsequently discretized and entropy-coded under a learned "prior". The approach shows a strong formal resemblance to VAEs and shall be reviewed in this terminology.

Let $\mathbf{z}$ be a continuous latent variable and $\hat{\mathbf{z}} = \lfloor\mathbf{z}\rceil$ the corresponding rounded, integer vector. The VAE-based compression approach consists of a stochastic encoder $e(\mathbf{z}|\mathbf{x})$, a prior $p(\mathbf{z})$, and a decoder $p(\mathbf{x}|\mathbf{z})$. The model is trained using the negative modified ELBO,

$$\mathcal{L}(\lambda, \mathbf{x}) = \mathbb{E}_{\mathbf{z}\sim e(\mathbf{z}|\mathbf{x})}[-\log p(\mathbf{x}|\mathbf{z}) - \lambda \log p(\mathbf{z})]. \quad (3)$$

The first term measures the average reconstruction of the data (distortion), while the second term measures the costs of entropy coding the latent variables under the prior (bitrate). The encoder $e(\mathbf{z}|\mathbf{x}) = \mathcal{U}(\text{Enc}_\phi(\mathbf{x}) - \frac{1}{2}, \text{Enc}_\phi(\mathbf{x}) + \frac{1}{2})$ is a boxed-shaped distribution that simulates rounding at training time through noise injection due to the reparameterization trick. Note it has zero entropy.

Once the VAE is trained, we en/decode data using the deterministic components as $\hat{\mathbf{z}} = \lfloor\text{Enc}(\mathbf{x})\rceil$ and $\hat{\mathbf{x}} = \text{Dec}(\hat{\mathbf{z}})$. We convert the continuous $p(\mathbf{z})$ into a discrete $P(\hat{\mathbf{z}})$ over the integer lattice as described in (Yang et al., 2022b) [Sec. 2.1.6] and use it for entropy coding $\hat{\mathbf{z}}$ (Ballé et al., 2018).

While VAE-based approaches have used simplistic (e.g., Gaussian) decoders, we show that can get significantly better results when defining the decoder $p(\mathbf{x}|\mathbf{z})$ as a conditional diffusion model.

## 3.2 CONDITIONAL DIFFUSION MODEL FOR COMPRESSION

The basis of our compression approach is a new latent variable model: the diffusion variational autoencoder. This model has a "semantic" latent variable $\mathbf{z}$ for encoding the image content, and a set of "texture" latent variables $\mathbf{x}_{1:N}$ describing residual information,

$$p(\mathbf{x}_{0:N}, \mathbf{z}) = p(\mathbf{x}_{0:N}|\mathbf{z})p(\mathbf{z}). \tag{4}$$

As detailed below, the decoder will follow a denoising process conditioned on $\mathbf{z}$. Drawing on methods described in Section 3.1, we use a neural encoder $e(\mathbf{z}|\mathbf{x}_0)$ to encode the image. The prior $p(\mathbf{z})$ is a two-level hierarchical prior (commonly used in learned image compression) and is used for entropy coding $\mathbf{z}$ after quantization (Ballé et al., 2018). Next, we discuss the novel decoder model.

**Decoder and training objective** We construct the conditional denoising diffusion model in a similar way to the non-variational diffusion autoencoder of Preechakul et al. (2022). In analogy to Eq. 1, we introduce a conditional denoising diffusion process for decoding the latent variables $\mathbf{z}$,

$$p_\theta(\mathbf{x}_{0:T}|\mathbf{z}) = p(\mathbf{x}_N) \prod p_\theta(\mathbf{x}_{n-1}|\mathbf{x}_n, \mathbf{z}) = p(\mathbf{x}_N) \prod \mathcal{N}(\mathbf{x}_{n-1}|M_\theta(\mathbf{x}_n, \mathbf{z}, n), \beta_n \mathbf{I}). \tag{5}$$

Since the texture latent variables $\mathbf{x}_{1:N}$ are not compressed but synthesized at decoding time, the optimal encoder and prior should be learned jointly with the decoder's marginal likelihood $p(\mathbf{x}_0|\mathbf{z}) = \int p(\mathbf{x}_{0:N}|\mathbf{z})d\mathbf{x}_{1:N}$ while targeting a certain tradeoff between rate and distortion specified by a Lagrange parameter $\lambda$. We can upper-bound this rate-distortion (R-D) objective as

$$\mathbb{E}_{\mathbf{z}\sim e(\mathbf{z}|\mathbf{x}_0)}[-\log p(\mathbf{x}_0|\mathbf{z}) - \lambda \log p(\mathbf{z})]$$

$$\leq \mathbb{E}_{\mathbf{z}\sim e(\mathbf{z}|\mathbf{x}_0)} \left[ \mathbb{E}_{\mathbf{x}_{1:N}\sim q(\mathbf{x}_{1:N}|\mathbf{x}_0)} \left[ -\log \frac{p(\mathbf{x}_{0:N}|\mathbf{z})}{q(\mathbf{x}_{1:N}|\mathbf{x}_0)} \right] - \lambda \log p(\mathbf{z}) \right]. \tag{6}$$

For brevity, we define $\log p_{\text{lower}}(\mathbf{x}_0|\mathbf{z}) = \mathbb{E}_{\mathbf{x}_{1:N}\sim q(\mathbf{x}_{1:N}|\mathbf{x}_0)} \left[ \log \frac{p(\mathbf{x}_{0:N}|\mathbf{z})}{q(\mathbf{x}_{1:N}|\mathbf{x}_0)} \right]$ as the variational lower bound to the diffusion model's conditional data likelihood (inducing an *upper* bound on the RD objective). We realize that $-\log p_{\text{lower}}(\mathbf{x}_0|\mathbf{z})$ corresponds to a novel *image distortion* metric induced by the conditional diffusion model (in analogy to how a Gaussian decoder induces the MSE distortion). This term measures the model's ability to reconstruct the image based on $\mathbf{z}$. In contrast, $\log p(\mathbf{z})$ measures the number of bits needed to compress $\mathbf{z}$ under the prior. As in most other works on neural image compression (Ballé et al., 2018; Minnen et al., 2018; Yang et al., 2022b), we use a box-shaped stochastic encoder $e(\mathbf{z}|\mathbf{x}_0)$ that simulates rounding by noise injection at training time.

Following Ho et al. (2020) and in analogy to Eq. 2, we simplify the training objective by using the denoising score matching loss,

$$-\log p_{\text{lower}}(\mathbf{x}_0|\mathbf{z}) \approx \mathbb{E}_{\mathbf{x}_0, n, \epsilon} ||\epsilon - \epsilon_\theta(\mathbf{x}_n(\mathbf{x}_0), \mathbf{z}, n/N_{train})||_\ell^\ell, \quad \ell = 1 \text{ or } \ell = 2. \tag{7}$$

The noise level $n$ and $\alpha_n$ are defined in Eq. 2. Instead of conditioning on $n$, we condition the model on the pseudo-continuous variable $n/N_{train}$ which offers additional flexibility in choosing the number of denoising steps for decoding (e.g., we can use a $N_{\text{test}}$ smaller than $N_{\text{train}}$). This pseudo-continuous scheme has a related continuous version (Kingma et al., 2021).

**Decoding process** Once the model is trained, we entropy-decode $\mathbf{z}$ using the prior $p(\mathbf{z})$ and conditionally decode the image $\mathbf{x}_0$ using ancestral sampling. We consider two decoding schemes: a stochastic one with $\mathbf{x}_N \sim \mathcal{N}(\mathbf{0}, \gamma^2 \mathbf{I})$ (where $0 < \gamma \leq 1$) and a deterministic version with $\mathbf{x}_N = \mathbf{0}$ (or $\gamma = 0$), both following the DDIM denoising method (Song et al., 2021a):

$$\mathbf{x}_{n-1} = \sqrt{\alpha_{n-1}} \left( \frac{\mathbf{x}_n - \sqrt{1-\alpha_n}\epsilon_\theta(\mathbf{x}_n, n, \mathbf{z})}{\sqrt{\alpha_n}} \right) + \sqrt{1-\alpha_{n-1}}\epsilon_\theta(\mathbf{x}_n, n, \mathbf{z}). \tag{8}$$

In both cases, the denoising network $\epsilon_\theta$ is defined as in Eq. 7. Since the latent variables $\mathbf{x}_{1:N}$ are not stored but generated at test time, these "texture" latent variables can result in variable reconstructions upon stochastic decoding (see Figure 4 for decoding with different $\gamma$ and Figure 14 for decoding the same image with different random seeds). We find that the DDPM scheme by (Ho et al., 2020) leads to worse results at decoding time (see Section 4.2 for details).

Algorithm 1 summarizes training and encoding/decoding. We find that the $\ell_1$ loss leads to better perceptual qualities and fewer color artifacts $\ell_2$, see also (Saharia et al., 2022).

---

**Algorithm 1:** Training (Left); Encoding and Decoding (Right). We note that the entropy coding part largely follows the method proposed by Ballé et al. (2018); the discrete latent variable $\hat{\mathbf{z}}$ is simulated by adding uniform noise during training and quantized to the nearest integer ($\hat{\mathbf{z}} = \lfloor \mathbf{z} \rceil$) during testing. We calculate discrete likelihood as $P(\hat{\mathbf{z}}) = CDF_p(\hat{\mathbf{z}} + 0.5) - CDF_p(\hat{\mathbf{z}} - 0.5)$ for a continuous distribution $p(\mathbf{z})$. Both $\lambda$ and $\rho$ are manually selected for training.

---

**while** *not converged* **do**

    Sample $\mathbf{x}_0 \sim$ dataset;

    $n \sim \mathcal{U}(0, 1, 2, .., N_{train})$;

    $\epsilon \sim \mathcal{N}(\mathbf{0}, \mathbf{I})$;

    $\bar{\mathbf{x}}_n = \sqrt{\alpha_n}\mathbf{x}_0 + \sqrt{1-\alpha_n}\epsilon$;

    $\hat{\mathbf{z}} \sim \mathcal{U}(\text{Enc}_\phi(\mathbf{x}_0) - \frac{1}{2}, \text{Enc}_\phi(\mathbf{x}_0) + \frac{1}{2})$;

    $\bar{\mathbf{x}}_0 = \frac{\bar{\mathbf{x}}_n - \sqrt{1-\alpha_n}\epsilon_\theta}{\sqrt{\alpha_n}}$;

    $L_{\mathrm{D}} = |\epsilon - \epsilon_\theta(\bar{\mathbf{x}}_n, i^n_{N_{\mathrm{train}}}\hat{\mathbf{z}})|$;

    $L = (1-\rho)L_{\mathrm{D}} + \rho d(\bar{\mathbf{x}}_0, \mathbf{x}_0) - \lambda \log_2 P(\hat{\mathbf{z}})$ ;

    $(\theta, \phi) = (\theta, \phi) - \nabla_{\theta,\phi}L$

**end**

Given $N_{test}$;

$\hat{\mathbf{z}} = \lfloor \text{Enc}_\phi(\mathbf{x}_0) \rceil$;

$\hat{\mathbf{z}} \xleftrightarrow{P(\hat{\mathbf{z}})}$ binary file;

$\bar{\mathbf{x}}_N = \mathbf{0}$ or $\mathbf{x}_N \sim \mathcal{N}(\mathbf{0}, \gamma^2\mathbf{I})$;

**for** *n=$N_{test}$ to 1* **do**

    $\epsilon_\theta = \epsilon_\theta(\bar{\mathbf{x}}_n, i^n_{N_{\mathrm{test}}}, \hat{\mathbf{z}})$;

    $\bar{\mathbf{x}}_0 = \frac{\bar{\mathbf{x}}_n - \sqrt{1-\alpha_n}\epsilon_\theta}{\sqrt{\alpha_n}}$;

    $\bar{\mathbf{x}}_{n-1} = \sqrt{\alpha_{n-1}}\bar{\mathbf{x}}_0 + \sqrt{1-\alpha_{n-1}}\epsilon_\theta$;

**end**

$\hat{\mathbf{x}}_0 = \bar{\mathbf{x}}_0$;

**return** $\hat{\mathbf{x}}_0$

---

**Optional Perceptual Distortion**      While Eq. 7 already describes a viable loss function for our conditional diffusion compression model, we can influence the perceptual quality of the compressed images by introducing additional loss functions similar to (Mentzer et al., 2020).

First, we note that the decoded data point can be understood as a function of the higher-level latent $\mathbf{x}_n$, the latent code $\mathbf{z}$, and the iteration $n$, such that $\bar{\mathbf{x}}_0(\mathbf{x}_n, \mathbf{z}, n) = \frac{\mathbf{x}_n - \sqrt{1-\alpha_n}\epsilon_\theta(\mathbf{x}_n, \mathbf{z}, i^n_N)}{\sqrt{\alpha_n}}$. When minimizing a perceptual metric $d(\cdot, \cdot)$ in image space, we can therefore add a new term to the loss:

$$L_{\text{perceptual}} = \mathbb{E}_{\mathbf{x}_0, \epsilon, n, \mathbf{z} \sim e(\mathbf{z}|\mathbf{x}_0)}[d(\bar{\mathbf{x}}_0(\mathbf{x}_n, \mathbf{z}, n), \mathbf{x}_0)]. \tag{9}$$

$$L = \rho L_{\text{perceptual}} - (1-\rho)\mathbb{E}_{\mathbf{z}\sim e(\mathbf{z}|\mathbf{x}_0)}[\log p_{\text{lower}}(\mathbf{x}_0|\mathbf{z}) + \frac{\lambda}{1-\rho}\log p(\mathbf{z})]. \tag{10}$$

This loss term is weighted by an additional Lagrange multiplier $\rho \in [0, 1)$, resulting in a three-way tradeoff between rate, distortion, and perceptual quality (Yang et al., 2022b; Blau & Michaeli, 2019). In this paper, we choose the widely-adopted LPIPS loss (Zhang et al., 2018a) for $L_{\text{perceptual}}$. (Note that Blau & Michaeli (2019) define the "perception" as a divergence of distributions without reference to the ground truth image, which is different from LPIPS.)

**Architecture**      The design of the denoising module follows a similar U-Net architecture used in DDIM (Song et al., 2021a) and DDPM (Ho et al., 2020) projects. Each U-Net unit includes two ResNet blocks (He et al., 2016), one attention block and a convolutional up/downsampling block. We use six U-Net units for both downsampling and upsampling process. The channel dimension for each downsampling unit is $64 \times j$, where $j$ is the index of the layer range from 1 to 6; the upsampling units follow the reverse order. Each encoder module consists of one ResNet block and one convolutional downsampling block. For conditioning with embedding, we use ResNet blocks and transposed convolution to upscale $\mathbf{z}$ to the same spatial dimension as the inputs of the beginning four U-Net downsampling units, so that we can perform conditioning by concatenating the the output of the embedder and the input of the corresponding U-Net unit. See Appendix B and Figure 5 for more details.

## 4   EXPERIMENTS

We conducted a large-scale compression evaluation involving 16 image quality metrics and 5 test datasets. Besides metrics measuring differences between compressed and raw images ("full reference metrics"), we also considered "no-reference metrics" that evaluate image quality without reference to the raw image. While some of these metrics are fixed, others are learned from data. We will refer to our approach as "Conditional Diffusion Compression" (CDC) in the following.

| PIEAPP(Prashnani et al., 2018) | Full Reference | Learned | Perceptual Metric |
|---|---|---|---|
| LPIPS(Zhang et al., 2018a) | Full Reference | Learned | Perceptual Metric |
| DISTS(Ding et al., 2020) | Full Reference | Learned | Perceptual Metric |
| CKDN(Zheng et al., 2021) | Full Reference | Learned | Perceptual Metric |
| MUSIQ(Ke et al., 2021) | No-Reference | Learned | Perceptual Metric |
| DBCNN(Zhang et al., 2018b) | No-Reference | Learned | Perceptual Metric |
| FID(Heusel et al., 2017) | No-Reference | Learned | Perceptual Metric |
| FSIM(Zhang et al., 2011) | Full Reference | Not Learned | Distortion Metric |
| SSIM(Wang et al., 2004) | Full Reference | Not Learned | Distortion Metric |
| MS-SSIM(Wang et al., 2003) | Full Reference | Not Learned | Distortion Metric |
| CW-SSIM(Sampat et al., 2009) | Full Reference | Not Learned | Distortion Metric |
| PSNR | Full Reference | Not Learned | Distortion Metric |
| GMSD(Xue et al., 2013) | Full Reference | Not Learned | Distortion Metric |
| NLPD(Laparra et al., 2016) | Full Reference | Not Learned | Distortion Metric |
| VSI(Zhang et al., 2014) | Full Reference | Not Learned | Distortion Metric |
| MAD(Larson & Chandler, 2010) | Full Reference | Not Learned | Distortion Metric |

Table 1: A list of the used evaluation metrics

**Metrics**   We selected 16 metrics from multiple categories: full-reference metrics, no-reference metrics, learned metrics, and not-learned metrics. We list these metrics and their corresponding categories in Table 1. Some more recently proposed learned metrics (Zhang et al., 2018a; Prashnani et al., 2018; Ding et al., 2020; Zheng et al., 2021) are believed to capture perceptual similarity better than other non-learned methods, we denote these metrics as *perceptual metrics* and the others as *distortion metrics*. Note that full reference metrics measure the differences between the compressed images and their corresponding ground truths, whereas no-reference metrics (except FID) calculate the scores without a specific ground truth. FID (Heusel et al., 2017) is a special no-reference metric since the scores are calculated by some *divergence* between compressed images and ground truth *distributions*. Similar to (Mentzer et al., 2020), we calculate FID by segmenting images into non-overlapping $256 \times 256$ resolution patches for small test sets ($\leq 100$ images).

**Test Data**   To support our compression quality assessment, we consider following datasets with necessary preprocessing: **1. Kodak** (Franzen, 2013): The data consists of 24 high-quality images at 768x512 (512x768) resolution. We do not evaluate the FID score of the dataset as 24 images only yield 144 image patches. **2. Tecnick** (Asuni & Giachetti, 2014): We use 100 natural images with 600x600 resolutions and we downsample these images to 512x512 resolution. **3. DIV2K** (Agustsson & Timofte, 2017): The validation set of this dataset contains 100 high-quality images. We resize the images with the shorter dimension being equal to 768px. Then, each image is center-cropped to a 768x768 squared shape. **4. COCO2017** (Lin et al., 2014): For this dataset, we extract all test images with resolutions higher than 512x512 and resize them to 384x384 resolution to remove compression artifacts. The resulting dataset consists of 2695 images. **5. ArtBench** (Liao et al., 2022): We use this dataset to conduct an out-of-distribution test, as it comprises 60000 images of artwork from 10 different artistic styles. We randomly select 1800 256x256 images from the *surrealism* style.

**Model Training**   We use the **Vimeo-90k** (Xue et al., 2019) dataset to train our model, consisting of 90,000 clips of 7-frame sequences at 448x256 resolution collected from vimeo.com. This dataset is widely used for video compression research. We randomly select one frame from each clip and crop the frame randomly to 256x256 resolution in each epoch. At the beginning of training, we warm-up the model by setting $\lambda = 10^{-4}$ and keep it running for around 500,000 steps. Then, we increase $\lambda$ to $\{0.0128, 0.0256, 0.0512\}$, respectively, and keep the model running for another 1,000,000 steps until the model converges. For the models with $\rho \neq 0$, we fine-tune the pretrained model with $\rho = 0$ for another 500,000 steps. We use $N_{\text{train}} = 20000$, batch_size=4 and the Adam (Kingma & Ba, 2014) optimizer in all cases. The learning rate is initialized as $lr = 5 \times 10^{-5}$ and then declines by 20% every 100,000 steps until $lr = 2 \times 10^{-5}$.

### 4.1 BASELINE COMPARISONS

**Baselines and Model Variants**   We showed two variants of our CDC model. Our first proposed model is optimized in the presence of an additive perceptual reconstruction term at $\rho = 0.9$. Here, we used DDIM stochastic decoding following $\mathbf{x}_N \sim \mathcal{N}(\mathbf{0}, \gamma^2 \mathbf{I})$ with $\gamma = 0.8$ to reconstruct the images. The other proposed version is the base model, trained without the additional perceptual term ($\rho = 0$) and using a deterministic decoding with $x_N = \mathbf{0}$. We used 500 iteration steps for

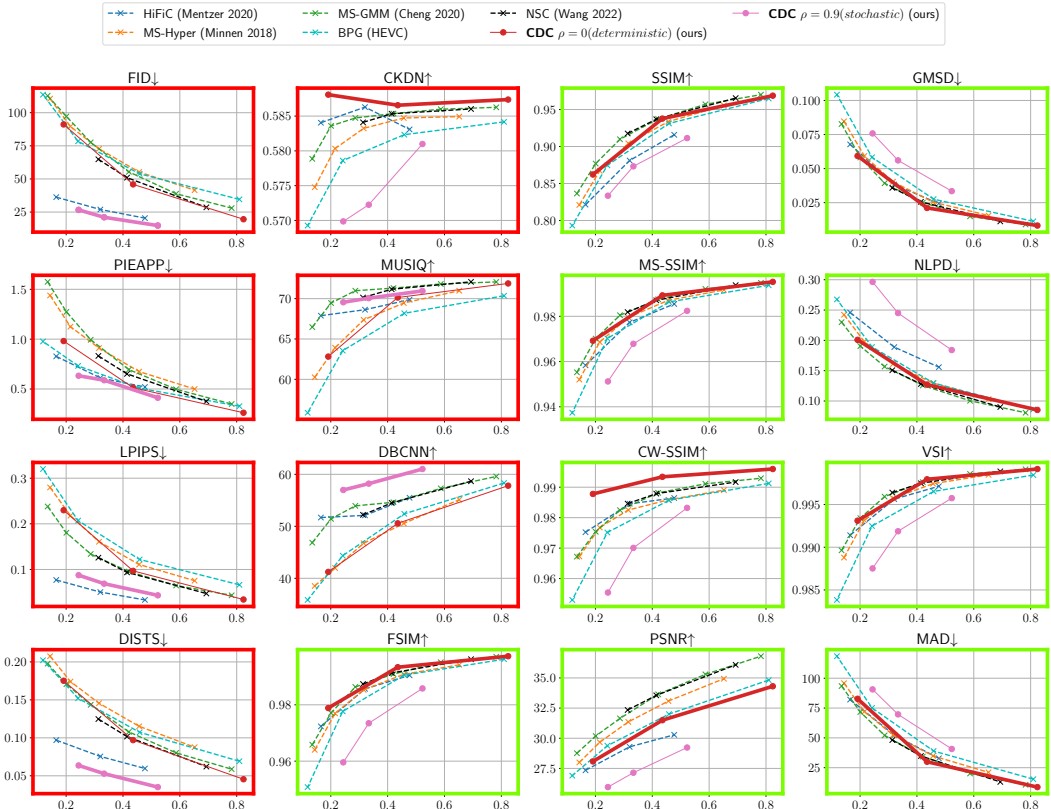

Figure 2: Tradeoffs between bitrate (x-axes, in bpp) and different metrics (y-axes) for various models tested on DIV2k (see Supplement G for other data sets). We consider both perceptual (red frames) and distortion metrics (green frames). Arrows in the plot titles indicate whether high (↑) or low (↓) values indicate a better score. As an overall tendency, we find that CDC (proposed) in its basic version (deterministic, without finetuning to LPIPS) compares favorably in distortion metrics, while CDC with stochastic decoding and added LPIPS losses performs favorably when focusing on perceptual metrics.

decoding. As discussed below, we find that this base version performs better in terms of distortion metrics, while the stochastic and LPIPS-informed version performs better in perceptual metrics.

We compare our method with several recent neural compression methods that are considered state of the art. The best reported perceptual results were obtained by **HiFiC** (Mentzer et al., 2020). This model is optimized by an adversarial network and employs additional perceptual and traditional distortion losses (LPIPS and MSE). In terms of rate-distortion performance, two current state-of-the-art models include **MS-GMM** (Cheng et al., 2020) and **NSC** (Wang et al., 2022). Both are improve over the MSE-trained Mean-Scale Hyperprior (**MS-Hyper**) architecture (Minnen et al., 2018). For comparisons with classical codecs, we also compare against the HEVC-based **BPG** codec.

Figure 2 shows the tradeoff between bitrates and image quality metrics on DIV2k dataset (see Supplement G for other data sets). Baseline models have dashed lines, and proposed ones (CDC) have solid lines. We will discuss subfigures according to their metric types, indicated by the frame color.

- **Perceptual Metrics (red)**. The red subplots contain perceptual metrics of Table 1. Cheng et al. (2020) and Wang et al. (2022) show high scores in MUSIQ, but show subleading performance in all other metrics. HiFiC is a much stronger baseline, but CDC($\rho = 0.9$) still outperforms all models in 4 out of 7 metrics. The fact that HiFiC performs better than CDC in LPIPS may be explained by the fact that CDC has access to both ground truth and reconstructed images at training time, while CDC only has access to stochastic estimates of the latter.

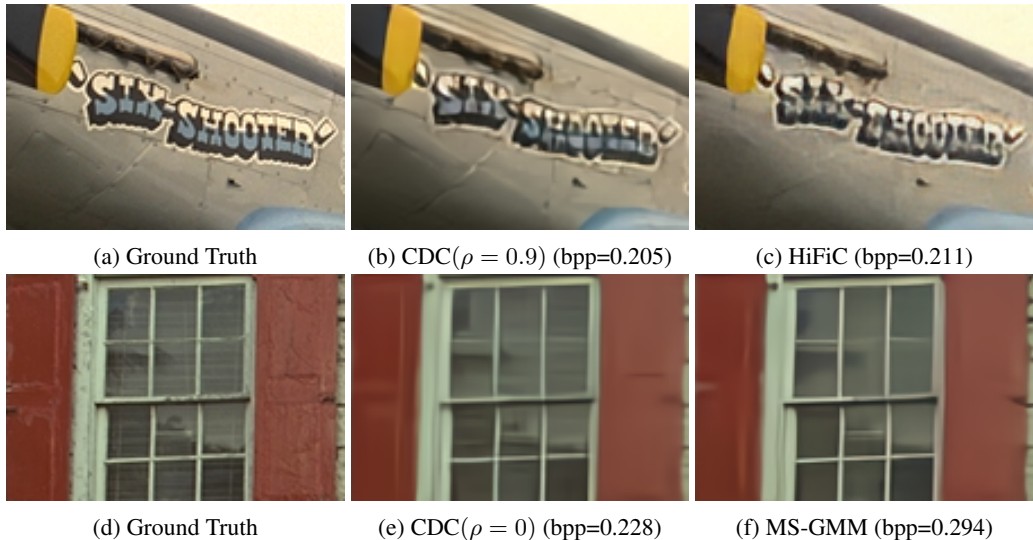

|  (a) Ground Truth | (b) CDC($\rho = 0.9$) (bpp=0.205) | (c) HiFiC (bpp=0.211) |
| (d) Ground Truth | (e) CDC($\rho = 0$) (bpp=0.228) | (f) MS-GMM (bpp=0.294) |

Figure 3: Reconstructed Kodak images at comparable bitrates. $1^{st}$ row: compared to HiFiC, our model more accurately retains the blue colors in the English letters and preserves more textures around the letters. $2^{nd}$ row: compared to MS-GMM, the diffusion model preserves more texture details (the curtain inside the window).

- **Distortion Metrics (green)**. Green subfigures include metrics listed as distortion-based in Table 1. "Cassical" neural compression models (Minnen et al., 2018; Wang et al., 2022; Cheng et al., 2020) directly target MSE distortion by minimizing an ELBO-objective with Gaussian decoders. In contrast, our approach is not optimized for this task. Surprisingly, our base model CDC($\rho = 0$) still gives relatively good results: it has an on-par distortion performance with the best baselines in 7 out of 9 metrics and the best score in CW-SSIM. As we expected, our model underperforms in PSNR score. Overall, we find that the deterministic version of our algorithm outperforms its stochastic variant in terms of distortion metrics.

It is widely acknowledged that current distortion-based metrics are not perfectly suited for assessing compression performance. For example, translating all pixels in an image by a single pixel leads to an unperceivable distortion, but a large MSE (Yang et al., 2022b). We therefore consider the first group of perception-based metrics more meaningful. In this aspect, our stochastic diffusion-based decoding scheme achieves very promising performance. Especially its strong performance in terms of FID, one of the most widely-adopted perceptual evaluation schemes (Ho et al., 2020; Song & Ermon, 2019; Mentzer et al., 2020; Brock et al., 2019; Song et al., 2021a;b), seems promising.

While the diffusion model's tendency to hallucinate high-frequency details may be visually appealing, it harms MSE distortion which encourages over-smoothing. This over-smoothing behavior is shown in Figure 3, where MS-GMM shows less details in the reconstruction compared to CDC.

## 4.2 STOCHASTIC DECODING

Our model allows both stochastic and deterministic decoding by varying the noise level $\gamma$ in the image decoding process. Since stochastic decoding is unintuitive and typically not desirable, one can make the decoding process still reproducible by using a fixed random seed. Additionally, we also consider DDPM (Ho et al., 2020) schemes that adds noise perturbation at every decoding step.

Figure 4 compares the compression performance of different decoding schemes in 4 perceptual and 4 traditional metrics (We note that the 8 metrics are sufficient to reflect the overall trend, so we omit the others due to the page limit). The plots show that larger $\gamma$ improves perceptual quality over the deterministic variants but bring down distortion scores when $\gamma \leq 0.8$. It also appears that the value $\gamma = 0.8$ yields the best perceptual scores and visualization quality, larger $\gamma$ not only leads to worse distortion but also perceptual quality. We also observe that DDPM cannot do better than the original DDIM scheme when using the same $\gamma$.

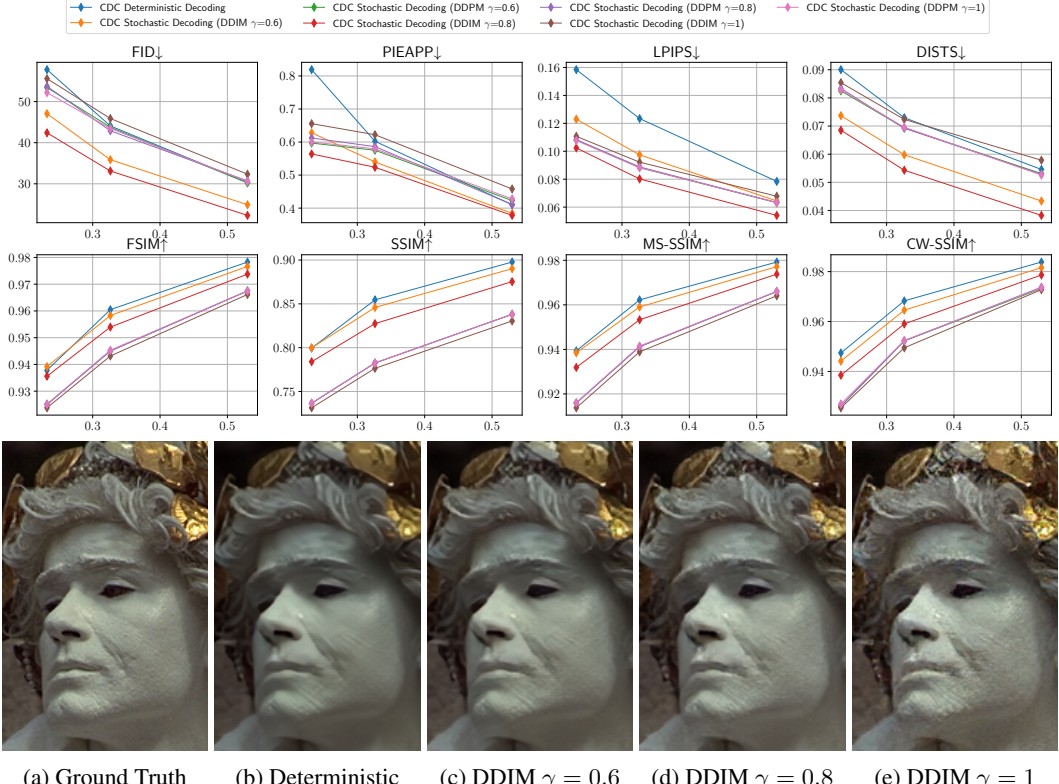

(a) Ground Truth    (b) Deterministic    (c) DDIM $\gamma = 0.6$    (d) DDIM $\gamma = 0.8$    (e) DDIM $\gamma = 1$

Figure 4: Quantitative (top figure) and qualitative (bottom figure) comparison of deterministic and stochastic decoding methods. Deterministic decoding typically results in a smoother image reconstruction. By increasing the noise $\gamma$ used upon decoding the images, we observe more and more detail and rugged texture on the face of the sculpture. Qualitatively, DDIM ($\gamma = 0.8$) seems to show the best agreement with the ground truth image.

## 5 CONCLUSION & DISCUSSION

We proposed a transform-coding-based neural image compression approach using diffusion models. We use a denoising decoder to iteratively reconstruct a compressed image encoded by an ordinary neural encoder. Our loss term is derived from first principles and combines rate-distortion variational autoencoders with denoising diffusion models. We conduct quantitative and qualitative experiments to compare our method against several state-of-the-art neural and classical codecs. Our approach achieves promising results in terms of the rate-perception tradeoff, outperforming several state-of-the-art baselines in four out of seven metrics, including FID. In terms of classical rate-distortion performance, our approach still performs comparable to most highly-competitive baselines.

**Future Research** It's a known issue that iterative decoding can be slow compared to a normal decoder. Whether or not this is relevant depends on the application, such as the available bandwidth for transmitting data. However, we can also trade-off decoding speed against image quality by varying the decoding steps or using recent ideas for accelerating diffusion models (Salimans & Ho, 2022). We also note that our work paves the path for future research on transform-coding based compression with diffusion models. Specifically, the compression performance can be further improved by incorporating the ideas from VAE-based coding methods such as more powerful prior distributions (Minnen et al., 2018; Minnen & Singh, 2020; Cheng et al., 2020), transformers (Zhu et al., 2021), iterative inference (Yang et al., 2020) and attention modules (Cheng et al., 2020) etc. Similarly, one can also improve the model from the diffusion model perspective, such as using different parameterizations (Ho et al., 2022a; Salimans & Ho, 2022), discrete or autoregressive diffusion distributions (Hoogeboom et al., 2021; Austin et al., 2021; Lee et al., 2021) and latent diffusion (Rombach et al., 2022).

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

## A PRETRAINED BASELINES

We refer to Bégaint et al. (2020) for pretrained MS-Hyper and MS-GMM models. For HiFiC model, we use the model implemented by a 3rd party researchers[1]. Both models were sufficiently trained on natural image datasets (Xue et al., 2019; Kuznetsova et al., 2020). For NSC (Wang et al., 2022) baseline, we use the official codebase[2] and DIV2k training dataset to train the model.

## B ARCHITECTURES

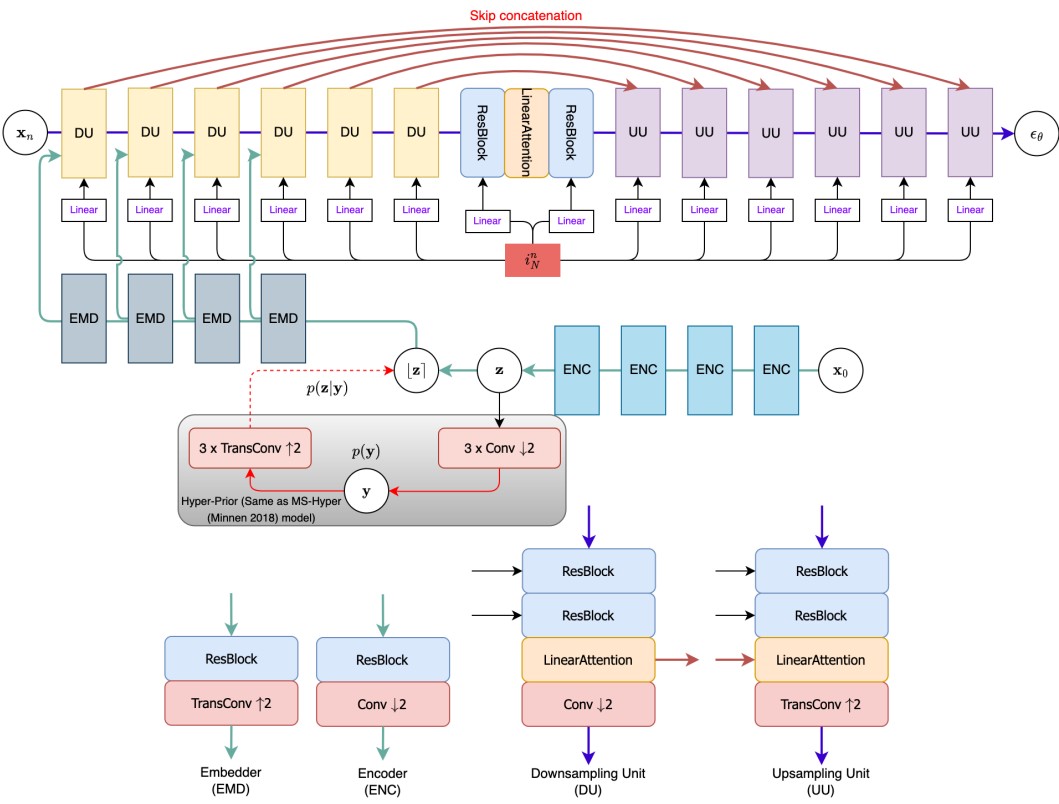

Figure 5: Visualization of our model architecture

Figure 5 describes our design choice of the model. We list the additional detailed specifications that we did not clarify in the main paper as follow:

- The hyper prior structure shares the same design as Minnen et al. (2018). The channel number of the hyper latent $\mathbf{y}$ is set as 256.

- We use 3x3 convolution for most of the convolutional layers. The only exceptions are the 1st conv-layer of the first DU component and the 1st layer of the 1st ENC component, where we use 7x7 convolution for wider receptive field.

- $i_N^n$ is embedded by a linear layer, which expand the 1-dimensional scalar to the same channel size as the corresponding DU/UU units. We then add the expanded tensor to the intermediate ResBlock of each DU/UU unit.

---

[1]https://github.com/Justin-Tan/high-fidelity-generative-compression
[2]https://github.com/Dezhao-Wang/Neural-Syntax-Code

## C    COMPUTE

We provide information on the model parameter size of the proposed model and baselines, and the corresponding time cost of running a full forward pass in Table 2. We run benchmarking on a server with a RTX Titan GPU. We execute each code snippet 24 times to encode and decode 24 images from Kodak dataset and calculate the average running time. We do not include entropy-coding time here as these models can share the same entropy-coding method.

|  | CDC (100 steps) | CDC (500 steps) | HiFiC | MS-GMM | NSC |
|---|---|---|---|---|---|
| Model Size (Mega Bytes) | 214.5 | 214.5 | 725.9 | 106.4 | 211.9 |
| Inference time (Seconds) | 10.91 | 54.92 | 1.01 | 0.52 | 0.86 |

Table 2: The memory usage and time consumption of a forward pass.

Our model also shows significantly better memory efficiency than HiFiC, but it is inevitable that our diffusion models have slow decoding speed due to the iterative denoising process. We note that the decoding speed is proportional to the number of decoding steps. As shown in Figure 6, decoding the image with 100 steps do not have observable qualitative difference to 500-step decoding, but it can be 5 times faster.

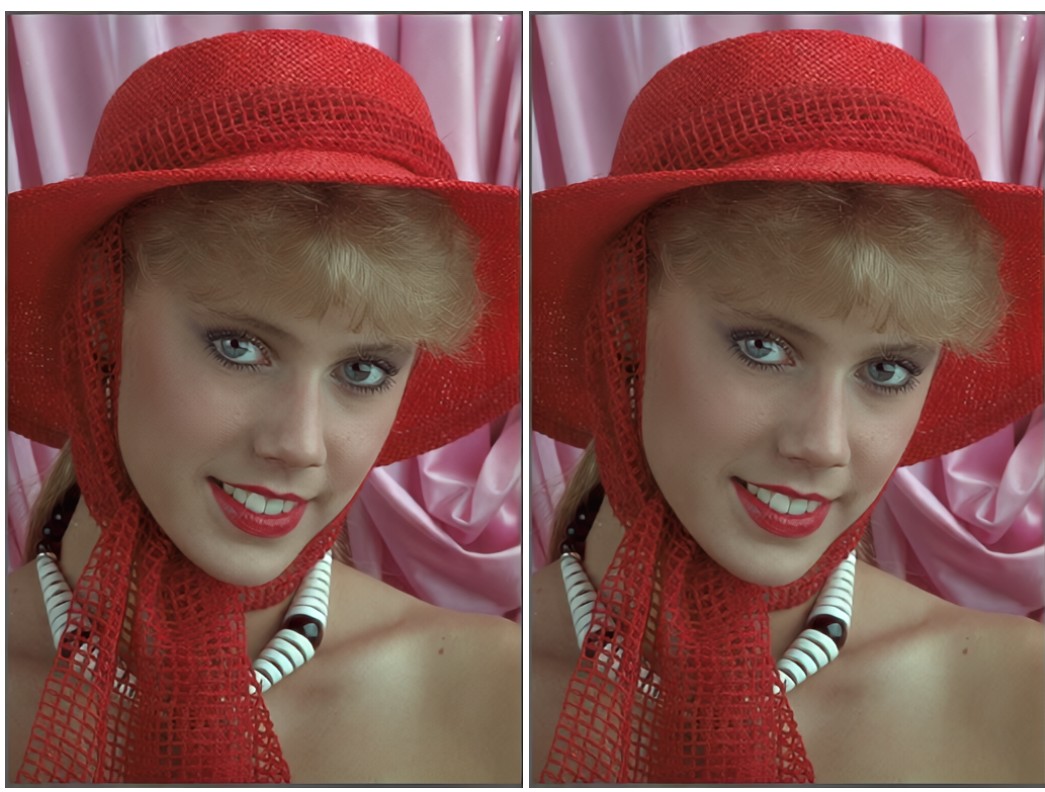

Figure 6: Qualitative comparison of models with different number of decoding steps. Left: 100 steps; Right: 500 steps

# D    SUPPLEMENTAL ABLATION STUDY

By varying the trade-off term $\rho$, we can train a model either prefer perceptual quality or traditional distortion performance. Figure 7 shows the rate-distortion curves for COCO dataset with deterministic decoding scheme. We consider four values in the study $(0, 0.32, 0.64, 0.9)$. The result shows that larger $\rho$ leads to better perceptual quality but worse distortions in most cases. We also note that $\rho > 0.9$ is not available as perceptual quality can not be perceivably improved and there is also a risk that the training may fail.

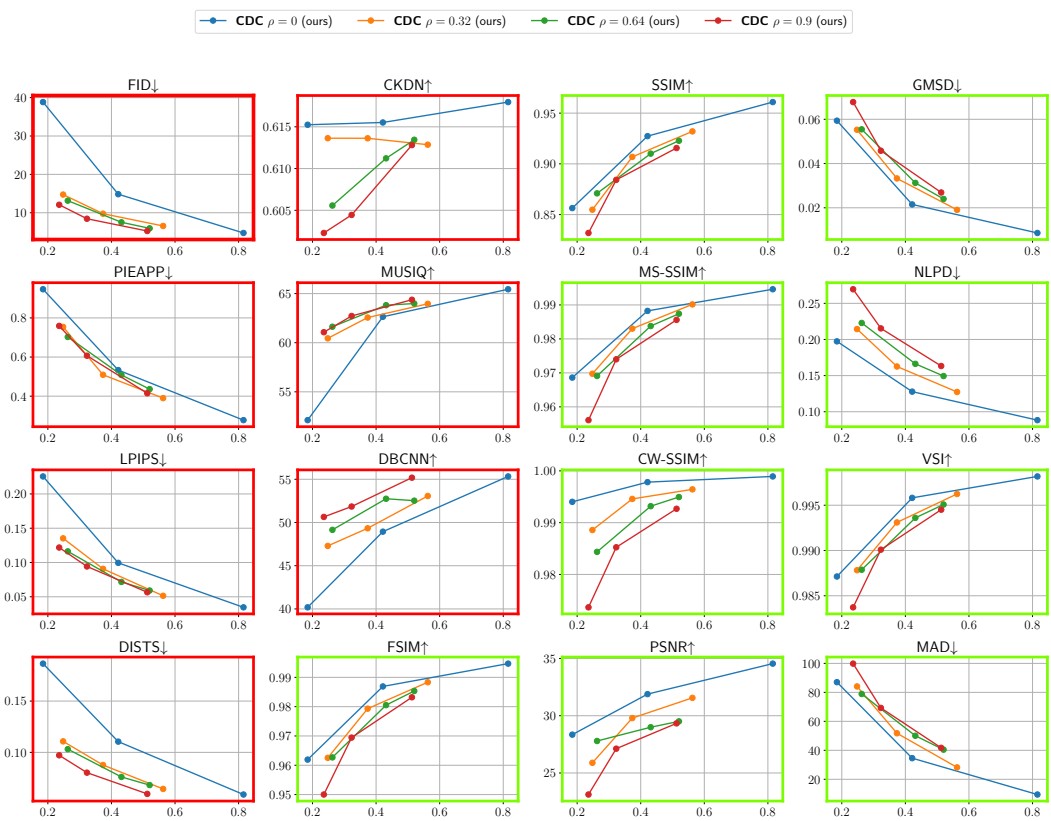

Figure 7: rate-distortion curves with different $\rho$ values

# E    ADDITIONAL VISUALIZATION OF THE COMPRESSED IMAGES

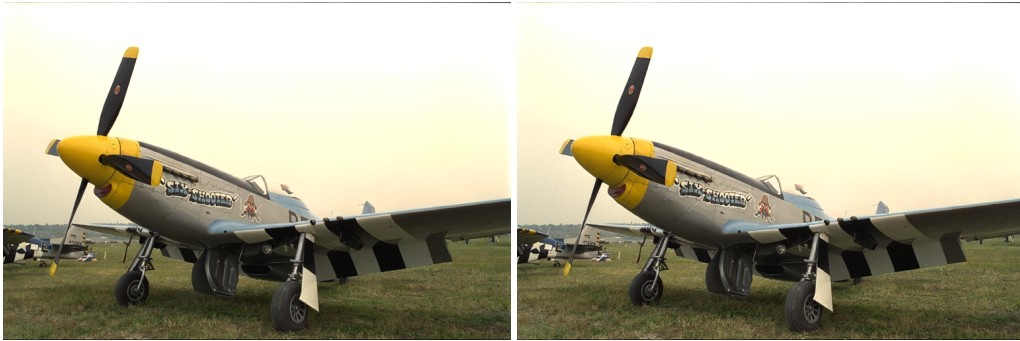

Figure 8: CDC($\rho = 0.9$), bpp=0.35. Left: deterministic; Right: stochastic

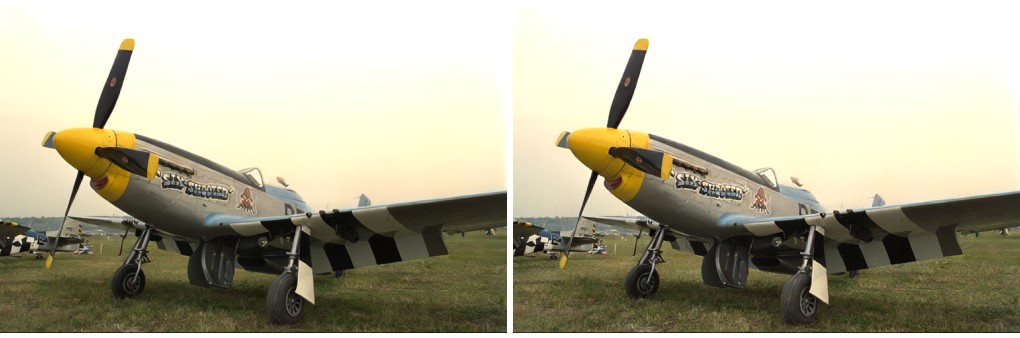

Figure 9: CDC($\rho = 0.9$), bpp=0.205. Left: deterministic; Right: stochastic

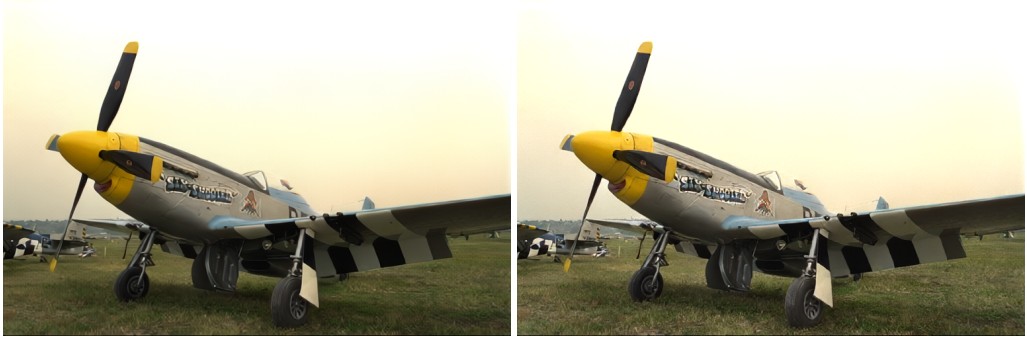

Figure 10: CDC($\rho = 0.9$), bpp=0.145. Left: deterministic; Right: stochastic

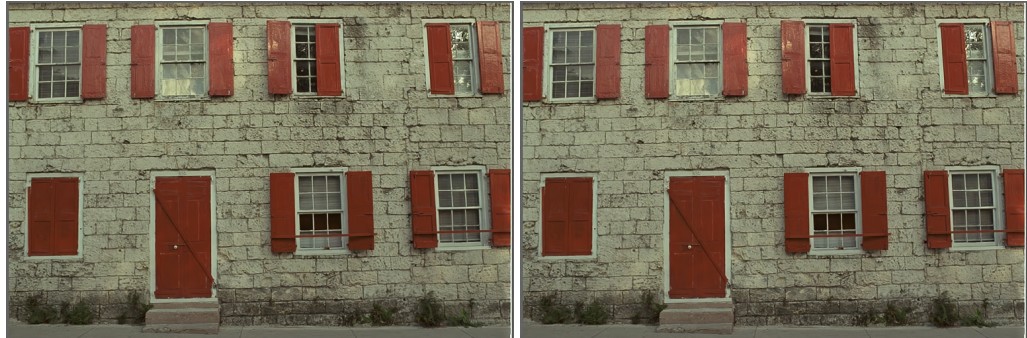

Figure 11: CDC($\rho = 0$), bpp=1.20. Left: deterministic; Right: stochastic

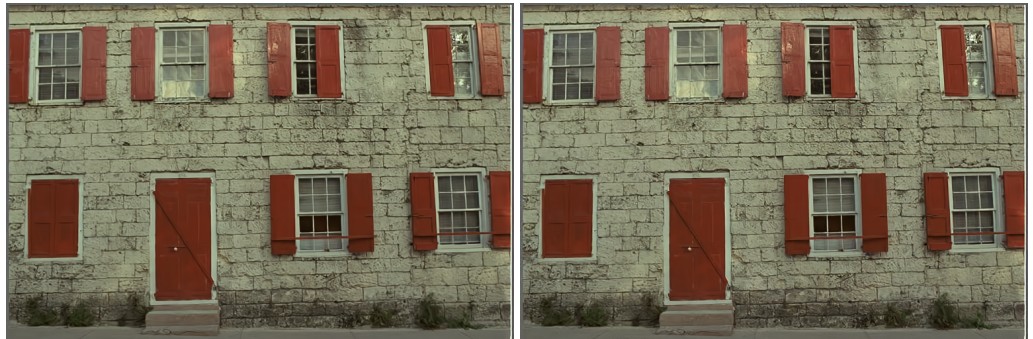

Figure 12: CDC($\rho = 0$), bpp=0.636. Left: deterministic; Right: stochastic

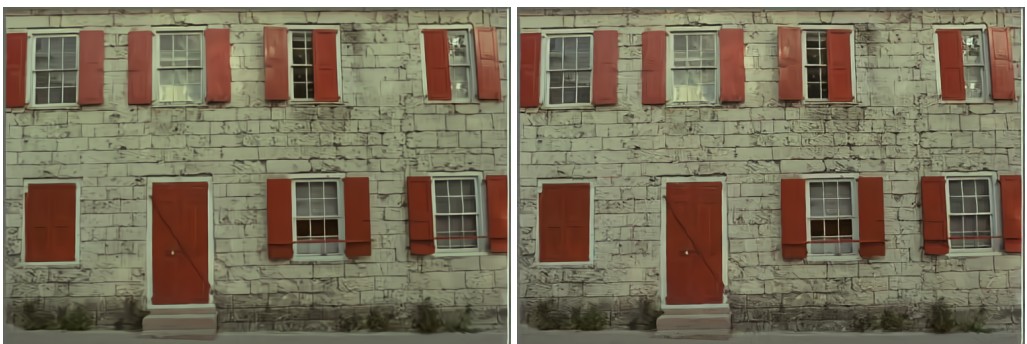

Figure 13: CDC($\rho = 0$), bpp=0.228. Left: deterministic; Right: stochastic

## F    DECODING VARIABILITY

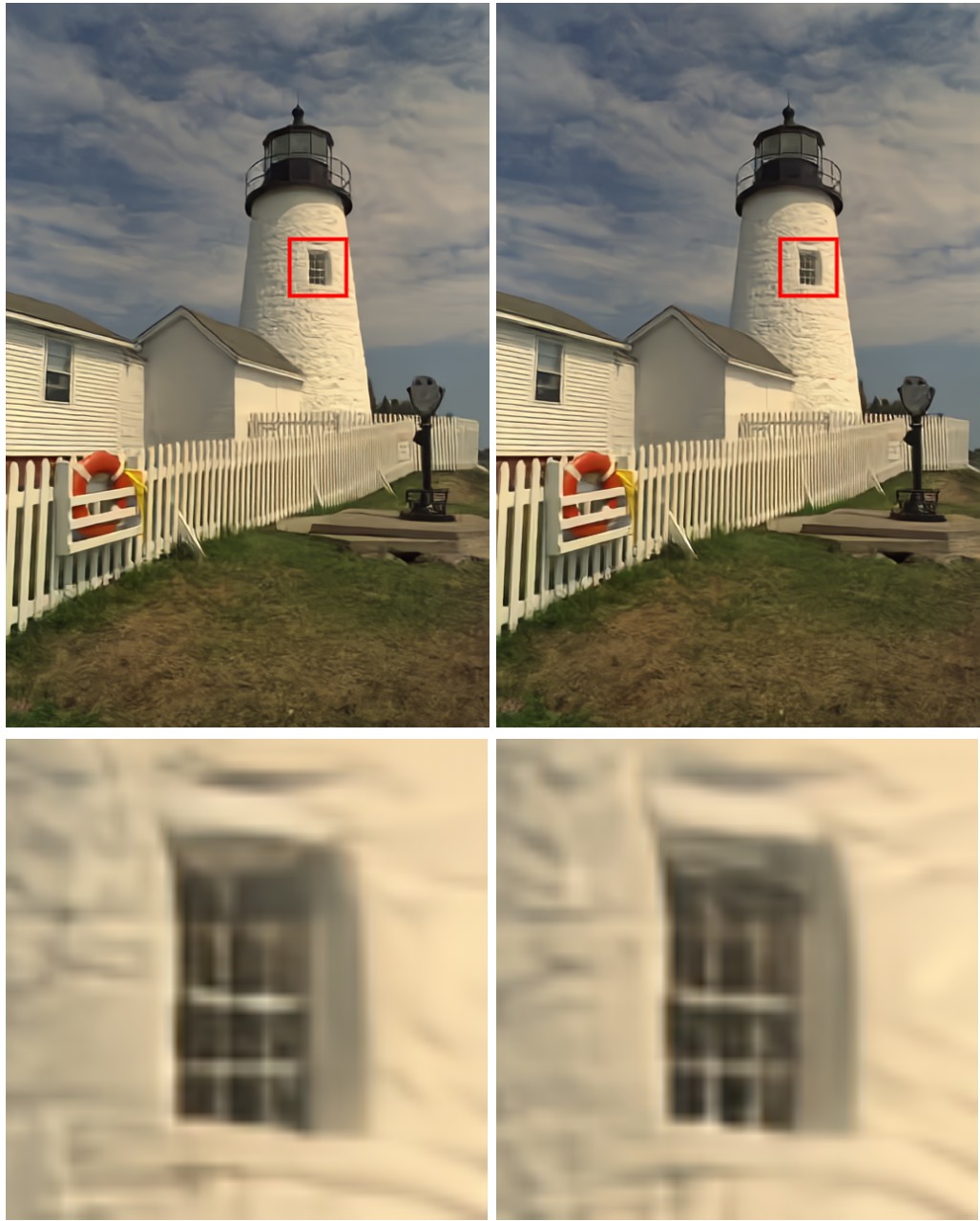

Figure 14: We stochastically decode the same latent variable $\mathbf{z}$ with the same $\gamma = 0.8$ but different random seed for $\mathbf{x}_N \sim \mathcal{N}(\mathbf{0}, \gamma^2 \mathbf{I})$. The decoded images only have small distinctions but we can still find there are slight detailed textural differences.

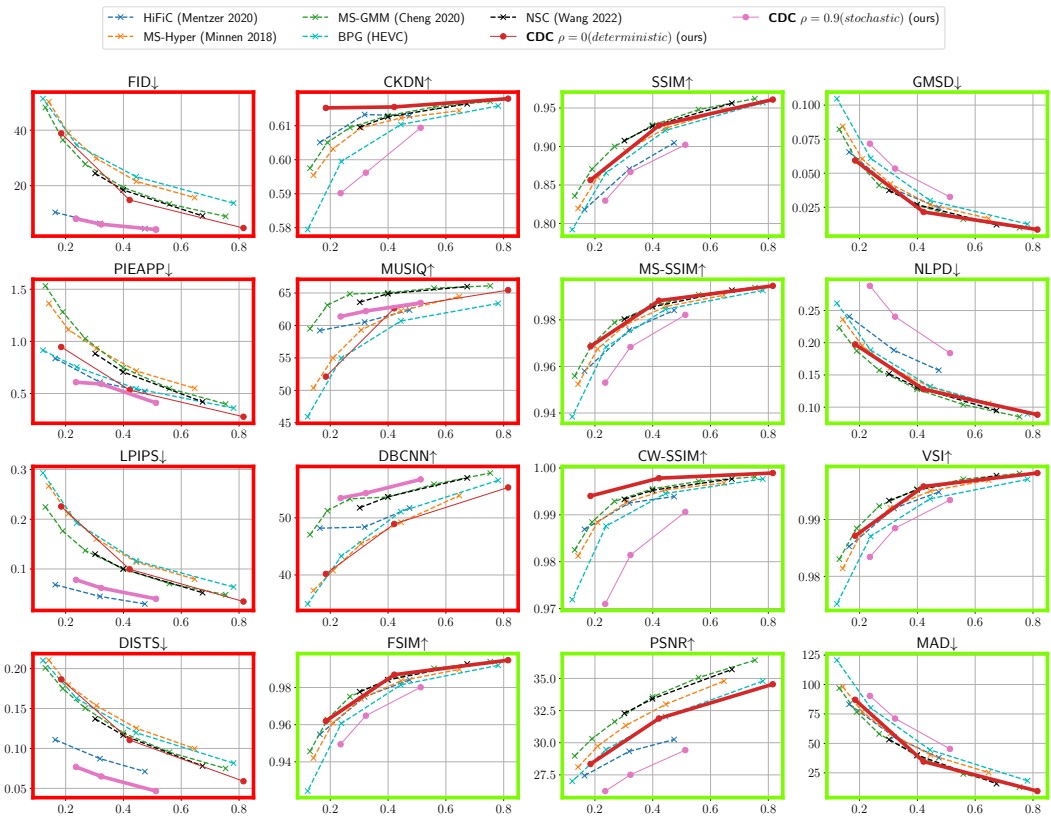

Figure 15: Rate-Distortion(Perception) for COCO dataset

# G ADDITIONAL RATE-DISTORTION(PERCEPTION) RESULTS

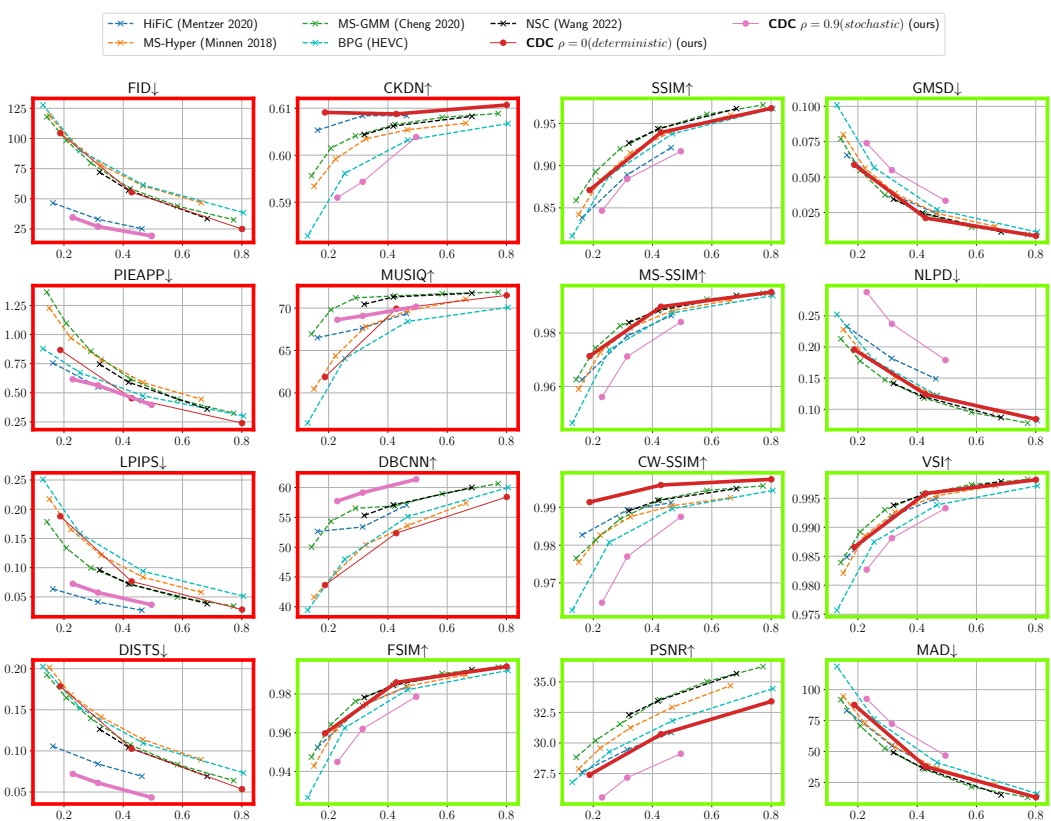

Figure 16: Rate-Distortion(Perception) for Tecnick dataset

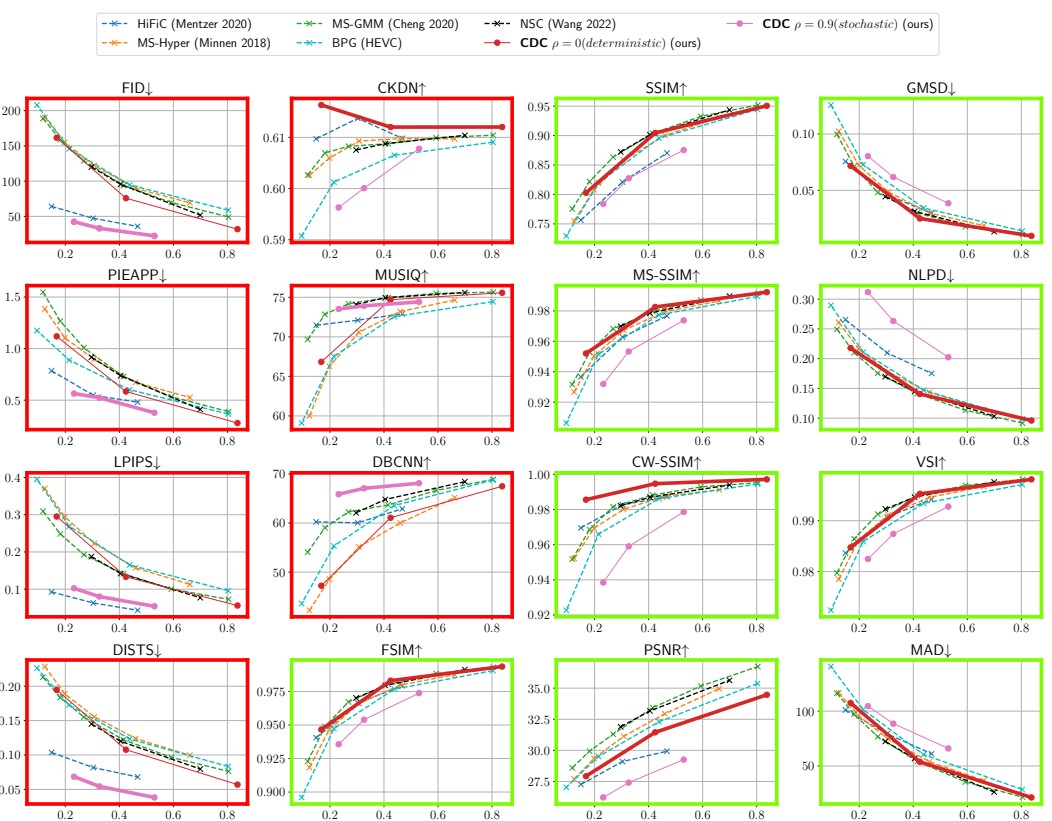

Figure 17: Rate-Distortion(Perception) for Kodak dataset

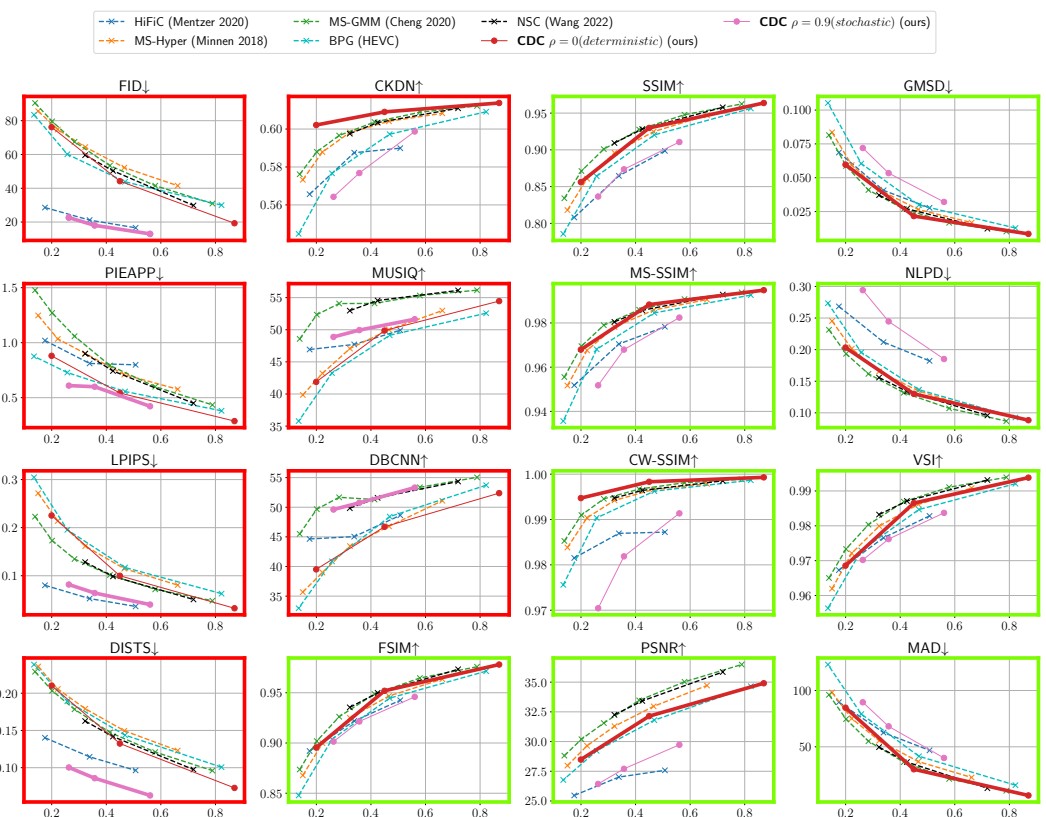

Figure 18: Rate-Distortion(Perception) for ArtBench(surrealism) dataset.

