# OpenReview forum: "Lossy Image Compression with Conditional Diffusion Models"
_ICLR.cc/2023/Conference — Submitted to ICLR 2023_

### Official Review · Reviewer_ocNf · 2022-10-17

**Confidence:** 3
**Correctness:** 3
**Technical Novelty And Significance:** 3
**Empirical Novelty And Significance:** Not applicable
**Recommendation:** 6

**Clarity, Quality, Novelty And Reproducibility:**

The paper is well and clearly written.  The novelty is a little limited, since it only simply introduces the diffusion model as encoder.

**Strength And Weaknesses:**

Strengths:
1. The idea is very intuitive and impressive. To enhance the capability of the decoder in image compression, it replace the Gaussian and Laplacian decoder with a powerful diffusion model.
2. The paper adopt as many as possible metrics (16) to evaluate the performance, which comprehensively reflects the advantages and disadvantages of the proposed method.

Weaknesses:
1. In this paper, $z$ is called semantic latent variable while $x_{1:N}$  texture latent variable. Why does $z$ and $x_{1:N}$ control the semantic and texture information respectively? If you could provide some visual proofs, it would be better.
2. The hyper-parameter $\rho$ makes a trade-off between the diffusion lower bound and the perceptual loss. More quantitative and qualitative comparisons under different $\rho$ values should be added to analyse its influence.
3. More visual results under different bitrates should be also provided.
4. The experimental comparisons are not sufficient enough, some recent works are not compared. E.g,
    Neural data-dependent transform for learned image compression, CVPR 2022.
    Enhanced invertible encoding for learned image compression, MM, 2021.
    Transformer-based transform coding, ICLR, 2021.


**Summary Of The Paper:**

This paper proposes a new image compression model based on the paradigm of neural image compression. The main novelty is to introduce a diffusion based decoder, instead of the traditional Gaussian or Laplacian decoders. Some experiments are conducted to verify the effectiveness.

**Summary Of The Review:**

In summary, the motivation of this work is clear, but it lacks some important analysis on the proposed method. Additionally, the experiments are insufficient.

---

> ### Author Response · Authors · 2022-11-16
> **Individual response**
>
> > In this paper, $z$ is called semantic latent variable while $x_{1:N}$ texture latent variable. Why does $z$ and $x_{1:N}$ control the semantic and texture information respectively?
>
> We thank you for your comment and added extra experiments. We hope Figure 4 and Figure 14 in appendix F give a strong motivation for this naming convention. Here, we demonstrated that stochastically decoding the same latent variable z multiple times (while repeatedly generating different “texture” latent variables $x_{1:N}$) results in the same image “content”, but varying local textures.
>
> > More analysis on the hyper-parameter $\rho$.
>
> We added an ablation study about \rho value in appendix D.  $\rho$ influences the perceptual quality; larger $\rho$ means higher perceptual quality and larger distortion. We find that $\rho=0.9$ is the largest value we can use in practice; larger $\rho$ will lead to unstable training and the perceptual quality performance improves only marginally.
>
> > More visual results under different bitrates
>
> Thank you for the suggestion. Appendix E shows additional visual results with 3 different bitrates and 2 decoding schemes.
>
> >More methods published after 2020
>
> Good suggestion. We added one baseline that you mentioned: Neural data-dependent transform for learned image compression, CVPR 2022 (NSC), which actually has similar performance to MS-GMM.

---

> > ### Comment · Reviewer_ocNf · 2022-11-28
> > **Feedback to authors**
> >
> > The response has addressed my issues. In general, the proposed method performs stably with respect to different metrics. I think it is a significant exploration in image compression based on diffusion model. Hence, I still remain my positive score.

---

> > > ### Author Response · Authors · 2022-11-29
> > > **Thank you**
> > >
> > > Thank you for your suggestion. Just a friendly reminder (as you may forget so): As far as what we can see, the score is currently "5: marginally below the acceptance threshold". If you appreciate our work, could you please improve the score accordingly? Thank you!

---

### Official Review · Reviewer_aJ8A · 2022-10-22

**Confidence:** 3
**Correctness:** 3
**Technical Novelty And Significance:** 2
**Empirical Novelty And Significance:** 2
**Recommendation:** 5

**Clarity, Quality, Novelty And Reproducibility:**

The quality of this paper is low. The motivation is not clear. Besides, the writing of the paper needs improvement. The paper is hard to understand.

**Strength And Weaknesses:**

Strength

+New attempt
This work tries to use the diffusion model for lossy image compression. As claimed, this is the first of this kind.

+Extensive experiment
To show the effectiveness of the proposed method, 5 datasets, and 16 IQA metrics are used. The proposed method achieves promising performance.

Weakness

-Unclear motivation
This is not clear why the use of the diffusion model is necessary. What is the advantage of the diffusion model for this task? To my best knowledge, the diffusion model is hard to preserve fidelity while fidelity is crucial for image compression. Besides, the diffusion model is slow, which also challenges the requirement of image compression.

-Limited novelty
It is intuitive to compress the image into the latent space. The diffusion model is a choice.  However, the novelty of how to use the diffusion model is limited in the work. All these ideas are intuitive or they have been used in previous related works.

-Insufficient analysis and comparison
1) The efficiency of different methods should be discussed.
2) Under different IQA metrics, the proposed with different parameter settings obtains better performance. How to choose the parameter when using the proposed method in practical applications？
3) More methods published after the 2020 year should be included for comparison.

-Other issues
1) It seems that the proposed method can only process the image with a size of 64X.
2) The ground truth image shown in the paper is not high-quality. It would be good if high-quality images can be used to show the advantages of the proposed method. For current visual comparison, it is hard to distinguish the performance of different methods as the ground truth images are still low-quality.
3) The paper is not well-written. Some content is hard to understand.


**Summary Of The Paper:**

This paper proposes a lossy image compression method using the condition diffusion model. This is an end-to-end framework based on a condition diffusion model. This is a new attempt to use the diffusion model for lossy image compression. Extensive experiments are conducted to show the advantages and robustness of the proposed method.

**Summary Of The Review:**

The paper has some significant novelty flaws and the technique and experiment are not convincing enough, which are the most important factors in my rating.

---

> ### Author Response · Authors · 2022-11-16
> **Individual response**
>
> > Unclear motivation This is not clear why the use of the diffusion model is necessary.
>
> We respectfully disagree that our motivation is unclear. Previous works already demonstrated that diffusion models can generate higher quality images than other generative models in various tasks. Diffusion models are therefore a natural candidate to explore in compression setups. In particular, we show enhanced performance in perceptual metrics (see the improved results in the paper).
>
> > Limited novelty.
>
> We respectfully disagree that being intuitive contradicts novelty. While transform-coding compression and conditional diffusion models are proposed works, the connection between both ideas remains unexplored. Other novel aspects of our work include the discussion of deterministic vs stochastic decoding, the ability to tune the model to a perceptual quantity of interest, and a solid experimental comparison against a variety of strong baselines and evaluation tasks.
>
> > Computational cost analysis
>
> Please see our general comment and we also added a discussion about computational efficiency in the paper appendix C
>
> > How to choose the parameter ($\rho$ and $\gamma$) when using the proposed method in practical applications
>
> Please check our Section 4.2 for an ablation study on stochastic decoding (post-processing) and Appendix D for an ablation study on $\rho$ values (pre-processing). Basically, stochastic decoding will increase the perceptual quality but worsen the distortion. We also note that the noise level should not be strong (the $\gamma$ value should be less than 0.8 in our case), otherwise the perceptual quality will also deteriorate.
>
> > More methods published after 2020
>
> Thanks for the suggestion. We added one baseline that was published in CVPR 2022 (NSC), which has similar performance to MS-GMM
>
> > It seems that the proposed method can only process the image with a size of 64X.
>
> We didn’t implement the code to support different resolutions as our testing data already has a size of 64X. In fact, we can add zero padding to the images with different resolutions to have a size of 64X. After decompression, we can remove the padded pixels to get the original size.
>
> > The ground truth image shown in the paper is not high-quality
>
> We show only a zoomed-in fragment of the high-resolution image in our paper to show the differences of the different compression methods. We now add the full images, which are shown in Appendix E.

---

### Official Review · Reviewer_Tnww · 2022-10-23

**Confidence:** 3
**Correctness:** 4
**Technical Novelty And Significance:** 2
**Empirical Novelty And Significance:** 2
**Recommendation:** 6

**Clarity, Quality, Novelty And Reproducibility:**

The paper is overall of decent quality, and is very well written. The proposed approach is well motivated and positioned in existing literature.

The novelty is relatively limited, as the method is a relatively straightforward extension of an existing neural image compression algorithm with a conditional diffusion model.

**Strength And Weaknesses:**

Strength

* Combining diffusion models and image compression is both interesting and expected. While previous works have explored this basic idea to a certain extent (as reviewed in this paper), this paper explored it most thoroughly to my knowledge. The comprehensive experiments provided valuable comparison between algorithms.

* The proposed perceptual loss is well placed in the diffusion model, which might find its use beyond the proposed approach. It is formally simple, and demonstrated to be able to substantially improve perceptual quality.

Weakness

* The paper rightly claimed their "competitive" performance against baseline models. However, the at best marginal improvement is hard to justify the significantly increased computational cost (e.g., 500 diffusion steps in decoding).

* The perceptual distortion loss could be better explored. E.g., It would be interesting to see how the rate-distortion trade-off is affected by different weights of the perceptual loss, but the paper only presented results from $\rho = 0$ and $\rho = 0.9$.

**Summary Of The Paper:**

This paper proposes a lossy image compression algorithm. The framework is similar to previous work of transform coding built on VAEs, and the proposed method replaced the the decoder with a conditional diffusion model. The paper provided comprehensive evaluations on various datasets and a number of metrics, and demonstrated that the proposed approach is compatible, and sometimes better than the state of the art.

**Summary Of The Review:**

The paper is well written, and provides valuable empirical comparison between the proposed and existing approaches on neural image compression. However, the novelty of this work is limited, and its improvement over baselines is marginal.

---

> ### Author Response · Authors · 2022-11-16
> **Individual response**
>
> > The at best marginal improvement is hard to justify the significantly increased computational cost
>
> We respectfully disagree for two reasons. First, while we agree that inference time can be a practical problem, diffusion-based compression approaches have significant advantages in terms of perceptual compression, as we demonstrate. In the discussion, we point out several options to further boost the inference speed by techniques such as distillation. Second, even at slow inference times, there are applications of extreme bandwidth or storage shortage where inference times may matter less. In any case, we feel that progress in engineering/science is often driven by imperfect but promising approaches that are picked up by others for further improvements.
>
> > The perceptual distortion loss trade-off weight $\rho$ could be better explored.
>
> We added an ablation study about $\rho$ value in appendix D.  $\rho$ influences the perceptual quality, larger $\rho$ means higher perceptual quality but larger distortion, but we find that $\rho=0.9$ is the largest value we can use since larger $\rho$ will lead to unstable training.

---

### Official Review · Reviewer_9ej8 · 2022-10-24

**Confidence:** 5
**Correctness:** 3
**Technical Novelty And Significance:** 2
**Empirical Novelty And Significance:** 2
**Recommendation:** 5

**Clarity, Quality, Novelty And Reproducibility:**

Clarity: The introduction and related work is very well written. However, this is not the case for the algorithm description. The reviewer finds it difficult to comprehend.

Quality and Novelty: Somewhat limited due to the limited rate-distortion performance and the ungiven computational complexity analysis.

Reproducibility: Based on the current description, the reviewer shall say it is less feasible to reproduce the results in the paper.

**Strength And Weaknesses:**

Strengths:

1. The use of conditional diffusion models in the context of neural image compression is somewhat novel.

2. The experimental results, especially using more than 15 image quality metrics, are comprehensive.

Weaknesses:

1. The proposed method is not clearly described.

a) Why the authors choose to start the derivation from the negative data log-likelihood (the inequality above Eq. (7)), which is more relevant to lossless image compression?

b) When and how do the index $n$ and $\mathbf{x}_n$ come into play in Eq. (9), provided that there are no such terms on the LHS?

c) Why is $\log p(\mathbf{z})$ instead of $\log P(\mathbf{\hat{z}})$ in the rate-distortion objective? Are we supposed to compute the discrete entropy as a proxy of the expected code length?

d) How to learn the quantization centers and how to model $\log P(\mathbf{\hat{z}})$ are not mentioned.

e) How to set the trade-off parameter $\lambda$ is not given. At lower bits, do we also need to reduce the dimensionality of the latent representation to encourage bit savings?

2. The rate-distortion performance is not outstanding compared to existing methods, e.g., HiFiC and MS-GMM, under standard and widely used metrics - PSNR and SSIM. Of particular interest, the proposed method jointly optimized for LPIPS (with $\rho=0.9$) under-performs HiFiC under the same metric - LPIPS. The authors may want to give some explanations.

3. The computational complexity in terms of the number of model parameters, and the encoding and decoding time should be provided to gauge whether the improved rate-distortion performance is worthwhile.

**Summary Of The Paper:**

This paper examines the use of conditional diffusion models (at the decoder side) for end-to-end optimized lossy image compression.

**Summary Of The Review:**

With a much clearer algorithm description, improved rate-distortion performance, and computational complexity analysis, this paper has a chance to get in.

---

> ### Author Response · Authors · 2022-11-16
> **Individual response**
>
> > Why the authors choose to start the derivation from the negative data log-likelihood
>
> We agree that deriving the RD-objective from the data likelihood was misleading. Our new derivation results from an upper-bound to the diffusion model’s marginal  rate-distortion objective (where the texture latent variables are marginalized out).
>
> > When and how do the index $n$ and $x_n$ come into play
>
> We apologize for the lack of clarity and revise the paper accordingly. We refer to equation 2 for the explanation of the two terms. $n$ is drawn from a uniform distribution. $x_n(n_0)$ is the noise-perturbed version of the image, as it is commonly defined in the context of diffusion models (Ho et al., 2020). We added explanations below the equation.
>
> > Why is $p(z)$ instead of $P(\hat z)$ in the rate-distortion objective
>
> We revised the section on “neural image compression” to make this more clear. In learned compression, we frequently need to convert probability densities over continuous spaces to discrete probability *distributions* over discrete spaces (after quantization). This follows standard practices explained in the revised paper version. In short, obtain the discrete likelihood of the integer z with the following formula: $P(z)=CDF_p(z+0.5)-CDF_p(z-0.5)$. We added a more detailed description in Algorithm 1.
>
> > How to learn the quantization centers
>
> We follow standard practice in learned compression, see [Yang et al., An Introduction to Neural Data Compression, 2022]. It is quite common to quantizing a learned distribution by rounding to an integer grid. During training, the rounding operation is simulated by adding a uniform noise. We improved the paper and Algorithm 1 accordingly.
>
> > How to set the trade-off parameter..
>
> lambda is a hyper-parameter that controls rate and distortion. There is no unique parameter lambda that performs best in all cases; lambda is sweeped over a large range to span the compression algorithm’s rate-distortion curve. Larger lambda means lower rate (file size) at larger distortion. In the experiment section, we mentioned that we considered $\lambda$ = (0.0128, 0.0256, 0.0512) for training. We did not lower the dimension to achieve lower bitrates and all the model shares the same dimensionality.
>
> > The rate-distortion performance is not outstanding
>
> Please see our general comment and our updated paper version. By experimenting with our stochastic version of the algorithm more, we obtained significantly better perceptual results. The fact that HiFiC performs better than CDC in LPIPS maybe explained by the fact that CDC has access to both ground truth and reconstructed images at training time, while CDC only has access to stochastic estimates of the latter. For SSIM, according to Figure 2, we think the performance is on-par with the baseline models (Ours may look worse because we only drew 3 data points). For your concern about PSNR and LPIPS, we revised our baseline analysis section and hope it can help explain your concern.
>
> > Computational complexity analysis
>
> Please see our general comment. We also measured the inference time and added a discussion about computational efficiency in the paper appendix C

---

### Official Review · Reviewer_NPK1 · 2022-10-24

**Confidence:** 3
**Correctness:** 3
**Technical Novelty And Significance:** 2
**Empirical Novelty And Significance:** 3
**Recommendation:** 5

**Clarity, Quality, Novelty And Reproducibility:**

**Clarity:** While the authors present an interesting and useful architecture, I think the paper could be clearer in terms of its presentation. The authors are doing their work a disservice by not presenting it more clearly, and I find that this detracts from the overall quality of the paper. Some of these include:

1. It is currently unclear whether a VAE decoder is used in the architecture (e.g. during training) or not. Judging from the loss function in eq. 7, a VAE decoder is not used. If so, the authors should clarify this because in Section 3.1, under "neural image compression", they make explicit reference to the VAE decoder.

2. Where exactly does the parameterisation in eq. 2 (parameterising a network which predicts the noise used to generate an image) fit within eq. 7? Presumably, the loss in eq. 2 is used to re-write the first term in the inner expectation in eq. 7 (which the authors refer to as $\log p_{\text{lower}}p(\mathbf{x}_0 | \mathbf{z})$) following Ho et al. 2020.

3. In Section 2, under "Diffusion models", the authors refer to relative entropy coding in a slightly misleading way. They state that "'relative entropy coding' (Flamich et al., 2020) is substantially slower than transform coding (Yang et al., 2022b; Balle et al., 2020)." However, this is not exactly the case because there exist several relative entropy coding algorithms such as those in Agustsson and Theis, 2020, Flamich et al. 2022 or Theis and Yosri, 2022 which render relative entropy coding computationally cheap. However, there does not (currently) seem to be an available method which is computationally fast without introducing additional coding costs or further restrictive assumptions.

**Quality:** Overall, the quality of the paper is good in terms of the technical contribution of the novel architecture introduced and the extensiveness of its empirical evaluation, however the quality of the writing could be improved. This includes improving the clarity of the paper, as well as addressing some of the grammar and syntax errors in the paper, which detract from its writing quality.

**Novelty:** While diffusion models have been used for the purposes of compression before, the architecture introduced by the authors appears novel. It is an interesting and useful architecture in the sense that it enables entropy coding to be used together with diffusion models and may therefore be of interest to the community.

**Reproducibility:** The authors provide several experimental details which appear sufficient to reproduce their results, although I have not examined this in detail. They provide two figures (Figures 1 and 5) explaining their architecture, and an outline of the algorithm (Algorithm 1) used for training and compression.


**References**

Jonathan Ho, Ajay Jain, and Pieter Abbeel. Denoising diffusion probabilistic models, 2020.

Eirikur Agustsson and Lucas Theis, Universally Quantized Neural Compression, 2020.

Gergely Flamich, Stratis Markou and Jose Miguel Hernandez Lobato, Fast Relative Entropy Coding with A* Coding, 2022.

Lucas Theis and Noureldin Yosri, Algorithms for the Communication of Samples, 2022.

**Strength And Weaknesses:**

## Strengths

**Novel image compression architecture:** The authors propose combining a VAE encoder together with a conditional diffusion model for performing image compression. This is a novel architecture which allows diffusion models to be used in conjunction with entropy coding. Specifically, the latent variable of the VAE can be coded using entropy coding, and the diffusion model can be conditioned on this latent variable to decode the image. This is an interesting idea since it enables high-performance diffusion models to be used with entropy coding, which is both fast and efficient (in terms of the code lengths it produces), rather than relying on relative entropy coding, which can be either prohibitively slow, restrictive in terms of its assumptions, or inefficient in terms of code length.

**Competitive performance and extensive evaluation:** The authors evaluate their architecture on several different test datasets, using a variety of evaluation metrics. On some of these metrics, their architecture outperforms the most competitive models which they evaluate, while on others it performs in par or markedly worse than many of the other models. While the proposed method does not outperform all competitive models across all metrics, the extensive evaluation which the authors conduct is very welcome. While they could have picked the subset of these metrics which is favourable towards their model, they have included a variety of metrics making the comparison more extensive and transparent.

## Weaknesses

**Significance of the contribution:** While the authors' architecture does exhibit performance competitive to that of several strong methods from the literature, it also performs markedly worse across several of the metrics used. As this is the sole contribution of the paper, it makes one wonder whether this is yet another compression architectures to the long list of existing methods. That being said, their architecture does enable entropy coding with a diffusion model as the decoder, and with further improvements this architecture could yield more promising results. However, I think that this work would benefit by addressing the following points:

1. The authors are advised to modify their assessment of how their model stacks up against alternatives. For example, in the conclusion they say "Our approach yields competitive rate-distortion(perception) performance against all baseline models", a claim which is not true since their model actually performs worse than state-of-the-art methods across several of the evaluation metrics. A more measured and nuanced position here (as well as throughout the paper) is necessary.
2. The authors could outline their ideas on further improving their architecture, to improve its compression performance. If, for example, the authors do not foresee any improvements on the performance that could be extracted from their architecture, then their contribution is of limited interest since it consists solely of this architecture and, presently, it is not entirely obvious that this architecture is clearly preferable than the alternatives they evaluate.
3. A more detailed discussion on the various evaluation metrics and their relationships to specific applications, would be welcome. For example, the authors could highlight the types of applications in which the evaluation metrics correlate with overall performance, and the kinds of settings in which their architecture might be more or less favourable.

**Summary Of The Paper:**

In this work, the authors propose a novel model architecture for lossy image compression. Specifically, they propose combining a VAE encoder architecture with a conditional diffusion decoder to compress image data. The authors derive an Evidence Lower Bound (ELBO) for their model, introducing a standard weighting coefficient for allowing different rate-distortion tradeoffs. They train their proposed model on the Vimeo-90k image dataset, and test it on five different image datasets, using a variety of sixteen different assessment metrics, showing promising results when compared with state-of-the art approaches.

**Summary Of The Review:**

In summary, I find the architecture proposed in this work an interesting and useful one. This architecture enables entropy coding to be used in conjunction with diffusion decoders, which is a good contribution to the literature, even though it does not exhibit the best performance across all the metrics used by the authors. The extensive evaluation done in the paper is also welcome.

However, I also found that the architecture was not particularly well explained, and the paper would benefit considerably from an effort to clarify how it operates, especially since this is the main contribution of this work. Currently, I would mark the paper as marginally below the acceptance threshold, giving it a score of 5, but encourage the authors to make an effort to improve their exposition, in which case I would be happy to consider improving this score. I also expect that the authors will adjust their claims regarding competitive performance across all evaluation metrics, which does not appear to be exactly the case.

---

> ### Author Response · Authors · 2022-11-16
> **Individual response**
>
>
> > The authors are advised to modify their assessment of how their model stacks up against alternatives.
>
> We agree and modified our assessment and discussion. Please check the general comments and paper for details.
>
> > The authors could outline their ideas on further improving their architecture.
>
> We agree and added an extended discussion at the end of the paper. Since our model is inspired by both traditional neural image-coding and diffusion models, one can incorporate ideas from both communities. For example, we can use more sophisticated entropy models (such as gaussian mixture priors, autoregressive priors, or iterative inference) or different diffusion-denoising processes (such as discrete diffusion or different parameterization of the prediction network). Also, please note that our experimental results improved significantly (see general comments above).
>
> > A more detailed discussion on the various evaluation metrics [...] would be welcome.
>
> We significantly simplified and revised the discussion of our method, baselines, and the involved metrics at the end of the baseline comparison section.
>
> >It is currently unclear whether a VAE decoder is used
>
> We have updated our method section. No VAE decoder is used in our architecture. We use a diffusion model to conditionally decode a latent state z, i.e., the diffusion model *replaces* a VAE decoder.
> > Where exactly does the parameterisation in eq. 2 (parameterising a network which predicts the noise used to generate an image) fit within eq. 6[..] Presumably, the loss in eq. 2 is used to rewrite the first term in the inner expectation in eq. 6
>
> This interpretation is correct and should be more clear now. We significantly rewrote and updated our method section, please check the new version. We derive our model essentially following Ho et al., 2020.
>
> >The authors refer to relative entropy coding in a slightly misleading way.
>
> We agree that relative entropy coding is an important line of research. For simplicity and without going into details, we revised the misleading sentences based on your suggestion.

---

### Author Response · Authors · 2022-11-16
**General Comments**

We thank all reviewers for their time and efforts. We tried our best to take their feedback into account by producing additional results and significantly revising our paper. Before responding to each reviewer individually, we would like to start with a few general comments.

**Method improvement**: most importantly, we discovered in the review period that a stochastic version of our method performs significantly better than the results reported in our earlier version. Our method titled “CDC $\rho=0.9$ stochastic” leads to better perceptual compression in a majority of metrics, especially surpassing HiFiC on the widely used FID score. Please find our new results in Figure 2.

**Paper improvement**: we significantly rewrote large parts of our paper, including the mathematical derivation of our loss function and the experiments section. We made our experimental results more accessible by summarizing our metrics by only two groups (perceptual metrics and distortion metrics) as opposed to three. It is now even more transparent that our stochastic decoding method performs significantly better than the deterministic version in terms of almost all perceptual metrics. Following reviewer suggestions, we added a recent baseline from 2022 to our experiments and added an extended discussion on our results and potential future model improvements.

**Additional experiments and shared concerns**: a shared concern was the long decoding time of our method. We ran additional experiments, measuring the runtimes in comparison to existing methods. We stress that, while diffusion models may be slower in practice than other approaches, this should not be the sole reason to discard research on the topic. Runtimes can be improved by recent techniques such as network distillation, which can significantly reduce the number of denoising steps (to less than 10 steps) without deteriorating image quality (Meng et al. 2022, Salimans et al. 2022). The cost analysis is available in appendix C.


Tim Salimans and Jonathan Ho. Progressive Distillation for Fast Sampling of Diffusion Models. In ICLR, 2022.

Chenlin Meng, Ruiqi Gao, Diederik P. Kingma, Stefano Ermon, Jonathan Ho, and Tim Salimans. On distillation of guided diffusion models. arXiv preprint, 2022.

---

### Decision · Program_Chairs · 2023-01-20

**Decision:**

Reject

**Justification For Why Not Higher Score:**

A majority of reviewers vote for rejection.

**Justification For Why Not Lower Score:**

N/A

**Metareview: Summary, Strengths And Weaknesses:**


Summary:

In this work, the authors propose a novel model architecture for lossy image compression. Specifically, they propose combining a VAE encoder architecture with a conditional diffusion decoder to compress image data. The authors derive an Evidence Lower Bound (ELBO) for their model, introducing a standard weighting coefficient for allowing different rate-distortion tradeoffs. They train their proposed model on the Vimeo-90k image dataset, and test it on five different image datasets, using a variety of sixteen different assessment metrics, showing promising results when compared with state-of-the art approaches.

Strengths:

- Novel image compression architecture
- Competitive performance and extensive evaluation
- The use of conditional diffusion models in the context of neural image compression is somewhat novel.
- The experimental results, especially using more than 15 image quality metrics, are comprehensive.
- Combining diffusion models and image compression is both interesting and expected.
- The proposed perceptual loss is well placed in the diffusion model,
- New attempt This work tries to use the diffusion model for lossy image compression. As claimed, this is the first of this kind.
- Extensive experiment To show the effectiveness of the proposed method, 5 datasets, and 16 IQA metrics are used. The proposed method achieves promising performance.
- The idea is very intuitive and impressive. T
- The paper adopt as many as possible metrics (16) to evaluate the performance.

Weaknesses:

- While the authors' architecture does exhibit performance competitive to that of several strong methods from the literature, it also performs markedly worse across several of the metrics used.
- the paper could be clearer in terms of its presentation.
- The proposed method is not clearly described.
- The rate-distortion performance is not outstanding compared to existing methods,
- The computational complexity in terms of the number of model parameters, and the encoding and decoding time should be provided to gauge whether the improved rate-distortion performance is worthwhile.
- The paper rightly claimed their "competitive" performance against baseline models. However, the at best marginal improvement is hard to justify the significantly increased computational cost
- The perceptual distortion loss could be better explored.
- Unclear motivation
- Insufficient analysis and comparison
- More visual results under different bitrates should be also provided.
- The experimental comparisons are not sufficient enough

Recommendation:

A majority of reviewers vote for rejection, so I therefore recommend rejection. I encourage the authors to look at the feedback provided and use this to improve the paper and resubmit to another venue.